# Amphiphysin I cleavage by asparagine endopeptidase leads to tau hyperphosphorylation and synaptic dysfunction

Xingyu Zhang[1†], Li Zou[1†], Lanxia Meng[1], Min Xiong[1], Lina Pan[1], Guiqin Chen[1,2], Yongfa Zheng[3], Jing Xiong[1,2], Zhihao Wang[2], Duc M Duong[4], Zhaohui Zhang[1], Xuebing Cao[5], Tao Wang[5], Li Tang[1], Keqiang Ye[2], Zhentao Zhang[1]*

[1]Department of Neurology, Renmin Hospital of Wuhan University, Wuhan, China; [2]Department of Pathology and Laboratory Medicine, Emory University School of Medicine, Atlanta, United States; [3]Department of Oncology, Renmin Hospital of Wuhan University, Wuhan, China; [4]Department of Biochemistry, Emory University School of Medicine, Atlanta, United States; [5]Department of Neurology, Union Hospital, Tongji Medical College, Huazhong University of Science and Technology, Wuhan, China

*For correspondence:
zhentaozhang@whu.edu.cn

†These authors contributed equally to this work

Competing interests: The authors declare that no competing interests exist.

**Abstract** Neurofibrillary tangles composed of hyperphosphorylated tau and synaptic dysfunction are characteristics of Alzheimer's disease (AD). However, the underlying molecular mechanisms remain poorly understood. Here, we identified Amphiphysin I mediates both tau phosphorylation and synaptic dysfunction in AD. Amphiphysin I is cleaved by a cysteine proteinase asparagine endopeptidase (AEP) at N278 in the brains of AD patients. The amount of AEP-generated N-terminal fragment of Amphiphysin I (1-278) is increased with aging. Amphiphysin I (1-278) inhibits clathrin-mediated endocytosis and induces synaptic dysfunction. Furthermore, Amphiphysin I (1-278) binds p35 and promotes its transition to p25, thus activates CDK5 and enhances tau hyperphosphorylation. Overexpression of Amphiphysin I (1-278) in the hippocampus of Tau P301S mice induces synaptic dysfunction, tau hyperphosphorylation, and cognitive deficits. However, overexpression of the N278A mutant Amphiphysin I, which resists the AEP-mediated cleavage, alleviates the pathological and behavioral defects. These findings suggest a mechanism of tau hyperphosphorylation and synaptic dysfunction in AD.

## Introduction

Alzheimer's disease (AD) is the most common neurodegenerative disease. Pathologically, it is characterized by the deposition of extracellular senile plaque and intracellular neurofibrillary tangles (NFTs). Senile plaques are composed of amyloid-β (Aβ), while NFTs are composed of hyperphosphorylated tau. The extent of tau deposition is closely related to the severity of cognitive impairments (*Arriagada et al., 1992*; *Brier et al., 2016*). Tau is expressed in neurons and mainly distributed in axons. It acts as a microtubule-associated protein and regulates the dynamics of microtubules, mediating neuronal polarity, axonal outgrowth, and axonal transport (*Dixit et al., 2008*; *Harada et al., 1994*; *Tan et al., 2019*). During the onset of AD, tau loses its physiological conformation and aggregates into amyloid fibrils. Aggregated tau further recruits monomeric tau and propagates in a 'prion-like' manner (*Frost et al., 2009*; *Clavaguera et al., 2009*). In the brain of AD patients, tau undergoes several posttranslational modifications including phosphorylation, truncation, acetylation, methylation, ubiquitylation, etc. (*Cook et al., 2014*; *Pevalova et al., 2006*).

Hyperphosphorylation is widely considered to be the major trigger that mediates tau pathology. Hyperphosphorylated tau loses its ability to associate with microtubules and is more prone to form paired helical filaments (PHFs) (*Alonso et al., 2001*; *Alonso et al., 1996*; *Moszczynski et al., 2015*). Cyclin-dependent kinase 5 (CDK5) is a major kinase involved in the abnormal phosphorylation of tau in AD brains (*Engmann and Giese, 2009*). The kinase activity of CDK5 is regulated by its regulatory subunit p35. P25, a truncated form of p35, is highly expressed in AD brains and constitutively activates CDK5 (*Patrick et al., 1999*; *Seo et al., 2014*). Thus, the activation of the p25/CDK5 pathway leads to the formation of neurofibrillary tangles and neurodegeneration (*Cruz et al., 2003*). But the mechanisms that regulate the production of p25 and the activation of CDK5 in AD remain poorly understood.

In addition to the deposition of Aβ and tau, synaptic dysfunction is an early and invariant feature of AD, occurring long before the onset of cognitive symptoms. The severity of dementia is correlated with the extent of synaptic loss (*DeKosky and Scheff, 1990*; *Terry et al., 1991*). It has been demonstrated that both Aβ and tau contribute to synaptic dysfunction. Soluble Aβ oligomers induce synaptic dysfunction and block hippocampal long-term potentiation via binding to receptors on neurons (*He et al., 2019*; *Laurén et al., 2009*; *Kim et al., 2013*), while tau is required for Aβ-induced synaptic dysfunction (*Ittner et al., 2010*). Several synaptic proteins are decreased in the brain tissue of patients with AD, while the concentration of some synaptic proteins in the cerebral spinal fluids is increased, indicating synaptic degeneration (*Bereczki et al., 2016*; *Bereczki et al., 2018*; *Kvartsberg et al., 2019*; *de Wilde et al., 2016*). Amphiphysin I is a synaptic protein with an N-Bin/amphiphysin/Rvs (BAR) domain, which is involved in clathrin-mediated endocytosis (*Lichte et al., 1992*). During the formation of newly retrieved presynaptic vesicles, Amphiphysin I senses and facilitates membrane curvature to mediate synaptic vesicles invagination and fission (*Takei et al., 1999*). Amphiphysin I also acts as a linker protein binding with dynamin, clathrin, Amphiphysin II, and other dephosphins in the clathrin-coated complex (*David et al., 1996*; *Takei et al., 1999*; *McMahon et al., 1997*; *Wigge et al., 1997*). Besides, Amphiphysin I also interacts with CDK5 activator p35 (*Floyd et al., 2001*). The expression of Amphiphysin I is reduced in brain regions with aggregates of hyperphosphorylated tau (*De Jesús-Cortés et al., 2012*), indicating dysfunction of Amphiphysin I is involved in the pathogenesis of AD. However, the relationship between Amphiphysin I dysregulation and tau phosphorylation during the development of AD is unknown.

Asparagine endopeptidase (AEP) is a cysteine protease that specifically cleaves the peptide bonds on the C-terminal side of asparagine residues (*Dall and Brandstetter, 2016*; *Chen et al., 1998*). Recently, we showed that AEP is activated in the aging brain, cleaves amyloid precursor protein (APP) and tau, promoting the deposition of Aβ and tau in the brain (*Zhang et al., 2015*). In this report, we show that AEP cleaves the synaptic protein Amphiphysin I after N278, generating Amphiphysin I (1-278) fragment. This fragment disrupts the normal endocytic function of Amphiphysin I, causing synaptic dysfunction. Furthermore, the Amphiphysin I (1-278) fragment promotes tau hyperphosphorylation through activating CDK5. Overexpression of Amphiphysin I (1-278) fragment in Tau P301S mice induces synaptic dysfunction and enhances tau hyperphosphorylation. Hence, we suggest that Amphiphysin I undergoes a novel posttranslational modification mediated by AEP, and the generated fragment promotes the pathogenesis of AD. Preventing the cleavage of Amphiphysin I by AEP is a therapeutic target for treating AD.

## Results

### AEP cleaves amphiphysin I in vitro

To investigate whether AEP cleaves Amphiphysin I in vitro, we incubated active AEP enzyme with GST-Amphiphysin I for 5 and 10 min, respectively. Amphiphysin I was cleaved into two fragments in a time-dependent manner (*Figure 1a*). To preclude the effect of other components in cell lysates, we purified the GST-Amphiphysin I and incubated it with AEP. Purified Amphiphysin I was also potently cleaved by active AEP enzyme (*Figure 1b*). Furthermore, we co-transfected GST-Amphiphysin I and myc-AEP into HEK293 cells. Wild-type AEP strongly induced Amphiphysin I fragmentation, while the AEP C189S mutant that abolishes its protease activity (*Li et al., 2003*) was unable to induce the cleavage of Amphiphysin I (*Figure 1c*). Moreover, the AEP inhibitor AENK suppressed

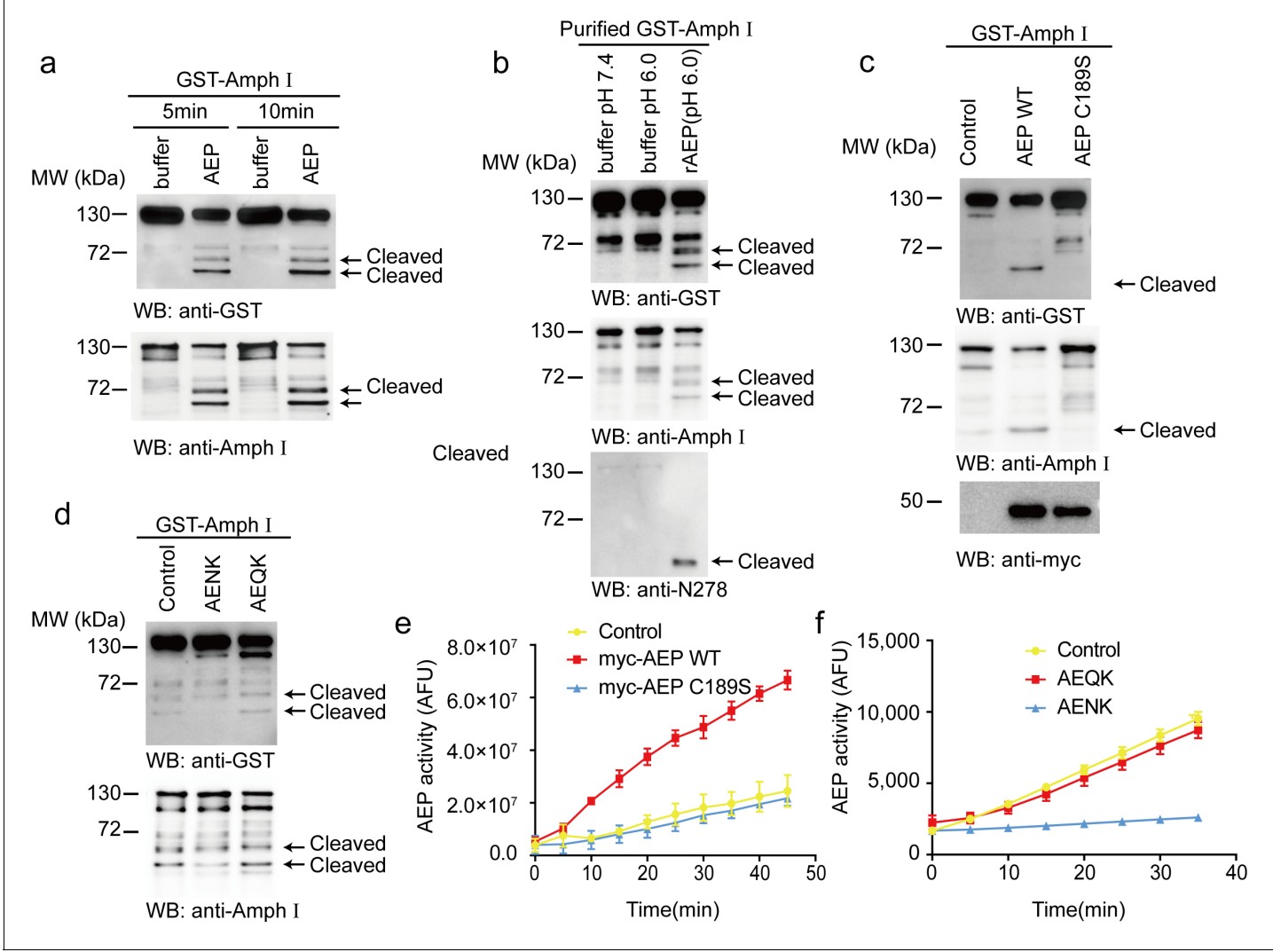

**Figure 1.** Asparagine endopeptidase (AEP) cleaves Amph I in vitro. (**a**) The cell lysate of HEK293 cells overexpressing GST-Amph I were incubated with AEP buffer (pH 6.0) or recombinant AEP for 5 min and 10 min, respectively. Western blot showing the cleavage of Amph I by AEP is increased in a time-dependent manner. (**b**) GST-Amph I expressing in HEK293 cells was purified by GST pull-down and incubated with AEP buffer (pH 7.4), AEP buffer (pH 6.0), or AEP in AEP buffer (pH 6.0) for 10 min. Western blot showing the cleavage of purified GST-Amph I recombinant protein by purified active recombinant AEP. (**c**) Cells co-transfected with GST-Amph I with myc-AEP WT or myc-AEP C189S. Western blot showing that only the myc-AEP WT cleaved GST-Amph I. (**d**) The lysates of cells overexpressing GST- Amph I were incubated with AEP and AENK (AEP inhibitor), AEQK. Western blot showing that AENK peptide but not AEQK inhibited the AEP activity. (**e**) AEP activity assay in the cell lysates transfected with WT and mutant AEP. Data represent mean ± s.e.m. of three independent experiments. (**f**) AEP activity assay verifying the inhibitory effect of AENK but not AEQK. Data represent mean ± s.e.m. of three independent experiments. MW, molecular weight; WB, western blot; GST, glutathione S-transferase. AFU, arbitrary fluorescence unit; DMSO, dimethyl sulfoxide.

The online version of this article includes the following source data for figure 1:

**Source data 1.** AEP activity assay in the cell lysates transfected with WT and mutant AEP.
**Source data 2.** AEP activity assay verifying the inhibitory effect of AENK but not AEQK.

the cleavage of Amphiphysin I, while its inactive analog AEQK did not affect this process (*Figure 1d*). The enzymatic activity of AEP was validated by fluorescent substrate cleavage assay (*Figure 1e,f*). Overall, these results indicate that Amphiphysin I is a substrate of AEP.

## AEP cleaves Amphiphysin I at N278 and generates the fragment 1–278 in AD

To confirm the cleavage of Amphiphysin I by AEP in vivo, we performed a mass spectrometry analysis of brain tissues from wild-type and AEP$^{-/-}$ mice. A (279-292) fragment of Amphiphysin I was found in wild-type mouse brain tissue lysate (*Figure 2a*). Comparative label-free proteomic analysis against AEP$^{-/-}$ mice revealed greater than ninefold enrichment for this fragment in wild-type versus AEP$^{-/-}$ brain extracts, indicating Amphiphysin I is cleaved by AEP after N278 (*Figure 2b*). To confirm the cleavage site of Amphiphysin I, we incubated the N278A mutant Amphiphysin I with AEP. The mutation completely blocked the production of the strongest fragment (*Figure 2c*), confirming that N278 is the major AEP cleavage site on Amphiphysin I.

Considering the mass spectrometry results in the mouse brain and the in vitro cleavage results, we focused on the Amphiphysin I (1-278) fragment. We generated an anti-Amphiphysin I N278 antibody which specifically recognizes the Amphiphysin I (1-278) fragment but not full-length Amphiphysin I or other Amphiphysin I fragments (*Figure 1b*, *Figure 2c*, *Figure 2—figure supplement 1a*). Using this antibody, we found that the Amphiphysin I (1-278) fragment was present and increased with age in brain lysates from wild-type mice and Tau P301S mice (*Figure 2—figure supplement 2a–d*). The expression of Amphiphysin I (1-278) fragment in AD brain tissues was higher than that in control brain tissues, as indicated by immunohistochemistry staining and Western blot (*Figure 2d*, *Figure 2—figure supplement 2e–f*). Furthermore, we found that Amphiphysin I (1-278) fragment colocalized with AT8 staining (*Figure 2e*). The demographic and pathological data of the patients are provided in *Supplementary file 1*. We also found the presence of Amphiphysin I N278 fragment in the cortex and hippocampal CA1 region from the Tau P301S transgenic AD mouse model, but few signals were found in the age-matched non-transgenic control brain sections (*Figure 2f*). Preincubation with Amphiphysin I (1-278) fragment abolished the immunoreactivity (*Figure 2—figure supplement 1b*), supporting the specificity of the anti-Amphiphysin I N278 antibody. These results indicate that AEP cleaves Amphiphysin I at N278 and generates the 1–278 fragment, which was enriched in the brain in an age-dependent manner.

## Amphiphysin I fragments induce synaptic dysfunction

Amphiphysin I interacts with other dephosphins such as clathrin, dynamin 1, and Amphiphysin II to regulate the invagination and fission of synaptic vesicles (*David et al., 1996*; *McMahon et al., 1997*; *Wigge et al., 1997*). To assess whether the fragmentation of Amphiphysin I influences its interaction with other dephosphins, we incubated GST-tagged full-length Amphiphysin I and AEP-derived N-terminal and C-terminal Amphiphysin I fragments (1–278 and 279–695) with mouse brain lysates, and performed GST pull-down assay. Clathrin and dynamin I bound to full-length Amphiphysin I and Amphiphysin I (279-695) fragment, while Amphiphysin II bound to full-length Amphiphysin I and the (1-278) fragment (*Figure 3a,b*). These results indicate that Amphiphysin I fragments generated by AEP partially lose their activity to bind other endocytic proteins, and may interfere with synaptic vesicles invagination and fission.

To investigate whether the Amphiphysin I fragments affect clathrin-mediated endocytosis, we overexpressed full-length Amphiphysin I or AEP-derived Amphiphysin I fragments (1–278 and 279–695) in COS-7 cells, respectively. Transferrin uptake assay found that transferrin uptake in cells transfected with Amphiphysin I (1-278) and Amphiphysin I (279-695) fragments was decreased by 25 and 22%, respectively, when compared with cells expressing full-length Amphiphysin I (*Figure 3c,g*). Furthermore, overexpression of active AEP fragment (26-323) also significantly inhibited transferrin uptake by 44% (*Figure 3d,h*). We further tested whether the presynaptic vesicle endocytosis in neurons was affected by the overexpression of Amphiphysin I fragments through FM 4–64 labeling assay. FM 4–64 labeling in boutons expressing Amphiphysin I (1-278) or Amphiphysin I (279-695) was inhibited by 37 and 64%, respectively, compared with that in boutons expressing full-length Amphiphysin I (*Figure 3e,i*). Besides, the density of dendritic spines in neurons expressing Amphiphysin I (1-278) or Amphiphysin I (279-695) decreased by 36 and 18%, respectively, compared with that in neurons expressing full-length Amphiphysin I (*Figure 3f,j*), indicating that Amphiphysin I fragments induce spine degeneration. All these results demonstrate that the Amphiphysin I fragments generated by AEP interfere with its function to interact with other dephosphins and impair the clathrin-mediated endocytosis and synaptic vesicle recycling.

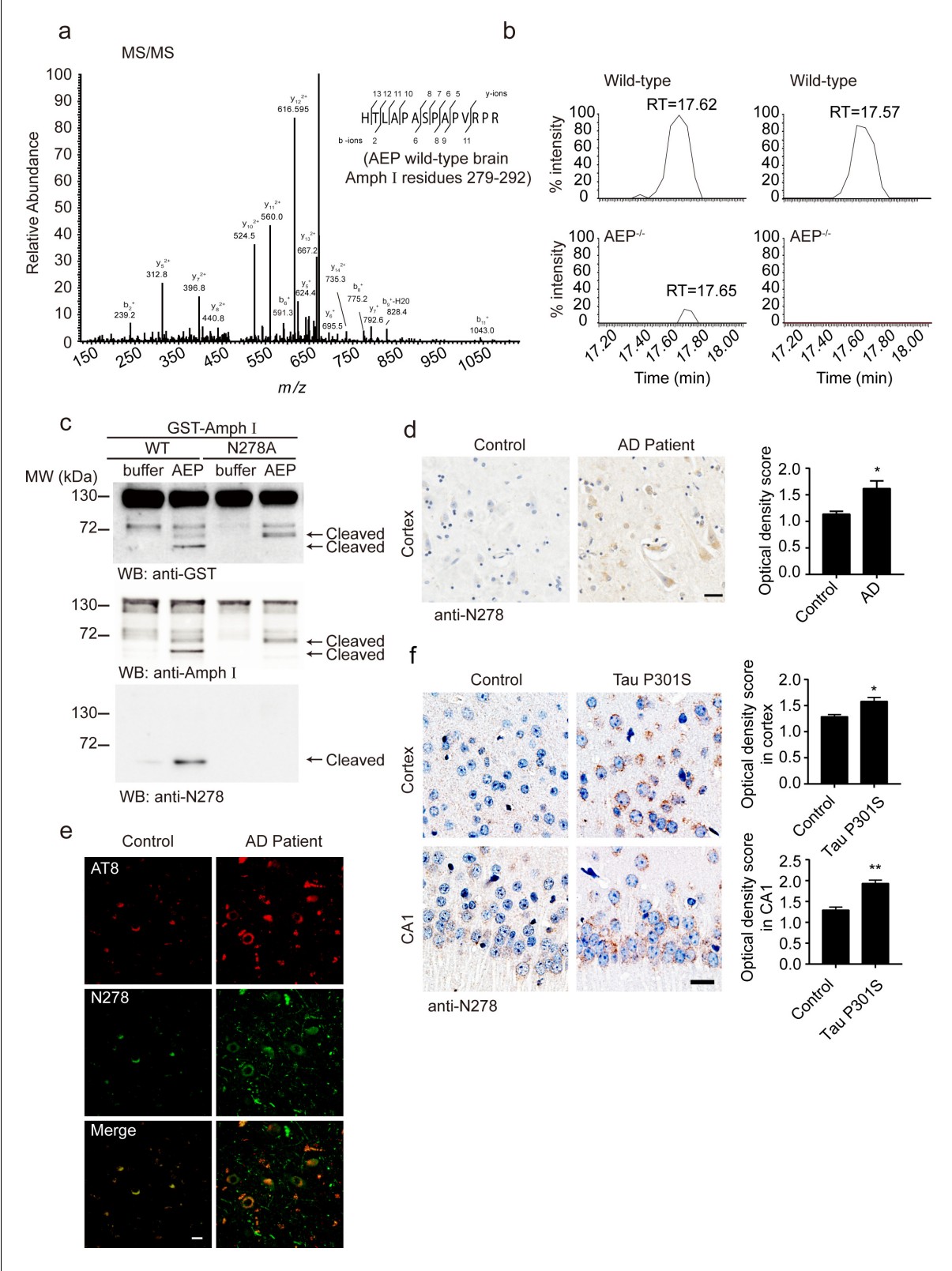

**Figure 2.** Asparagine endopeptidase (AEP) cleaves Amph I in N278 and generates the fragment 1–278 in Alzheimer's disease (AD). (**a**) MS/MS spectrum showing the cleavage of Amph I after N278 in AEP wild-type brain samples. (**b**) Representative extracted ion chromatograms for Amph I peptide from two wild-type and two AEP$^{-/-}$ mouse brain samples. Signal intensities were then normalized to wild-type samples, setting the maximum signal intensity to 100%. Values are represented as raw peptide extracted ion intensity. (**c**) Cleavage of mutant Amph I by AEP. Amph I cleavage was analyzed by

*Figure 2 continued on next page*

*Figure 2 continued*

Western blot after glutathione S-transferase (GST)-Amph I wild-type or N278A mutant was incubated with recombinant AEP. (**d**) Immunohistochemistry (IHC) of Amph I N278 fragments in brain sections from subjects with AD and control subjects (mean ± s.e.m.; Student's t-test, n = 4 mice per group; ***p<0.001). Scale bar, 20 μm. (**e**) The co-immunostaining of anti-N278 (green) with p-tau (red) in brain sections from subjects with AD and control subjects. Scale bar, 20 μm. (**f**) Immunohistochemistry (IHC) of Amph I N278 fragments in brain sections from 8-month-old Tau P301S mice and wild-type mice (mean ± s.e.m.; Student's t-test, n = 4 mice per group; ***p<0.001). Scale bar, 20 μm.

The online version of this article includes the following source data and figure supplement(s) for figure 2:

**Source data 1.** Immunohistochemistry (IHC) of Amph I N278 fragments in brain sections from subjects with AD and control subjects.

**Source data 2.** Immunohistochemistry (IHC) of Amph I N278 fragments in brain sections from 8-month-old tau P301S mice and wild-type mice.

**Figure supplement 1.** Verification of the anti-Amph I N278 antibody.

**Figure supplement 2.** The Amph I (1-278) fragment exists in wild-type, Tau P301S mouse and Alzheimer's disease (AD) patients.

**Figure supplement 2—source data 1.** Western blot analysis of Amph I, Amph I(1-278) and AEP in wild-type mouse brain during aging.

**Figure supplement 2—source data 2.** Western blot analysis of Amph I and Amph I(1-278) in tau P301S mouse brain during aging.

**Figure supplement 2—source data 3.** Western blot analysis of Amph I and Amph I (1-278) fragment in control and AD human brain samples.

To test whether Amphiphysin I fragmentation causes presynaptic defects, we recorded miniature excitatory autaptic currents (mEACs) from individually inhabited neurons grown on collagen-poly-D-lysine (PDL) islands (*Figure 4a*). The frequency but not the amplitude of mEPSCs was significantly decreased in neurons expressing Amphiphysin I (1-278) or Amphiphysin I (279-695) than that in neurons expressing full-length Amphiphysin I (*Figure 4b,c*). These results suggest that Amphiphysin I fragments interfere with synaptic transmission.

## Amphiphysin I fragments promote the hyperphosphorylation of tau

The hyperphosphorylation of tau is a major pathological marker of AD and is believed to mediate neurodegeneration. To investigate whether Amphiphysin I fragments affect tau hyperphosphorylation, we overexpressed EGFP, EGFP-tagged full-length Amphiphysin I, and AEP-derived Amphiphysin I fragments (1–278 and 279–695), respectively, in the hippocampus of Tau P301S transgenic mice. Two months later, we found a marked increase of AT8 and AT100 immunoreactivity (*Figure 4d*) in hippocampus sections overexpressing Amphiphysin I (1-278) compared with other groups. Amphiphysin I (279-596) fragment also slightly enhanced tau phosphorylation compared with full-length Amphiphysin I (*Figure 4d*). However, the levels of total tau were comparable among all groups, indicating that Amphiphysin I fragments promote tau phosphorylation but do not elevate the expression of total tau. We further confirmed the effect of Amphiphysin I fragments on tau phosphorylation in cultured primary neurons. We found increased tau phosphorylation in neurons overexpressing Amphiphysin I fragments (*Figure 4e,f*). The AT100 staining was mainly found in the nucleus, which is consistent with the previous reports (*Gil et al., 2017*; *Hernández-Ortega et al., 2016*, *Gärtner et al., 1998*). To confirm the specificity of the AT100 antibody, we stained brain slides from tau knockout mice and found the signals were completely abolished (*Figure 4—figure supplement 1*). These in vivo and in vitro results indicate that Amphiphysin I (1-278) enhances the phosphorylation of tau.

## Amphiphysin I (1-278) fragment enhances tau hyperphosphorylation through activating CDK5 kinase activity

CDK5 is one of the major kinases that induce the hyperphosphorylation of tau during AD pathology (*Zheng et al., 2005*). To figure out the mechanisms underlying tau hyperphosphorylation induced by Amphiphysin I (1-278) fragment, we assessed the activity of CDK5 in neurons infected with adeno-associated virus (AAV) encoding full-length Amphiphysin I and AEP-derived Amphiphysin I fragments, respectively (*Figure 5a*). CDK5 was immunoprecipitated and incubated with excessive CDK5 substrate, histone H1. After 30 min of incubation, phosphorylated histone H1 was detected by immunoblot (*Figure 5a*). We found that the activity of CDK5 in neurons expressing Amphiphysin I (1-278) was much higher than that in other groups (*Figure 5d,e*), indicating that Amphiphysin I (1-278) enhances the kinase activity of CDK5. Consistent with the increased CDK5 activity, the content of AT8 and AT100 was increased in neurons expressing Amphiphysin I (1-278) (*Figure 5b,c*). To confirm whether Amphiphysin I (1-278)-induced hyperphosphorylation of tau is mediated by CDK5, we treated the neurons with 10 μM CDK5 inhibitor roscovitine in neurons overexpressing Amphiphysin I

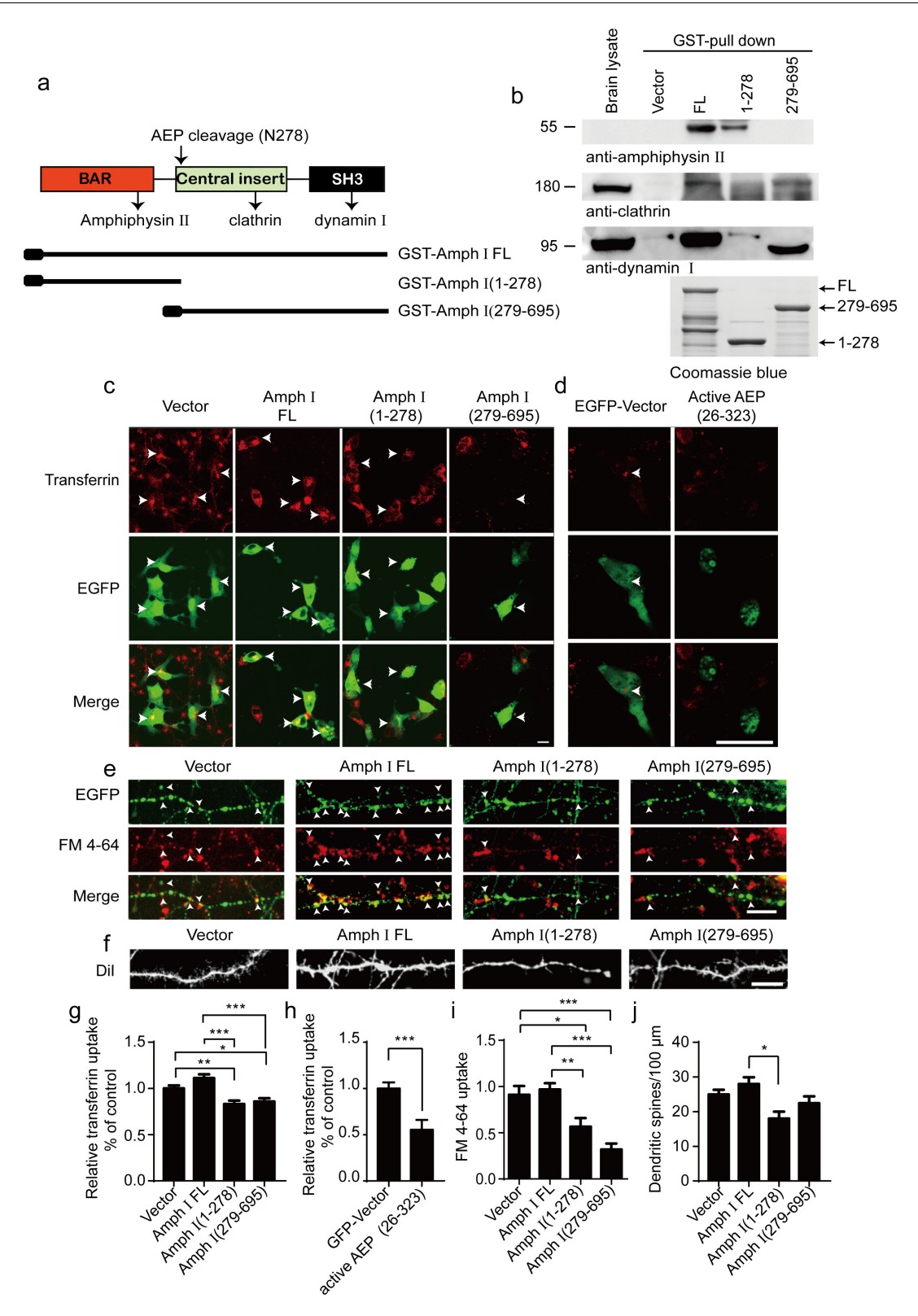

**Figure 3.** Amph I fragments disrupt clathrin-mediated endocytosis. (a) Schematic diagram of Amph I domains and its cleavage by asparagine endopeptidase (AEP). The binding sites of Amph I to other dephosphins are also shown. (b) Glutathione S-transferase (GST) pull-down assay. GST vector, GST-Amph I, GST-Amph I (1-278), or GST-Amph I (279-695) bound to glutathione agarose beads were incubated with mice brain lysate. The bound proteins were detected by anti-amphiphysin II, anti-clathrin, anti-dynamin I antibodies. Bottom: The Coomassie blue stain of bGST-Amph I,

*Figure 3 continued on next page*

*Figure 3 continued*

bGST-Amph I (1-278), or bGST-Amph I (279-695). (**c, d**) Transferrin uptake assay. Confocal laser scanning micrographs showing the uptake of transferrin by COS-7 cells expressing EGFP, EGFP-Amph I FL, EGFP-Amph I (1-278), EGFP-Amph I (279-695), and EGFP-active AEP (26-323), respectively. Top: Transferrin uptake in transfected cells; middle, EGFP expression in COS-7 cells; bottom, the merge of EGFP and transferrin uptake. The arrows indicate transferrin uptaken by transfected cells. Scale bar, 20 μm. (**e**) FM 4–64 uptake assay in neurons expressing EGFP, EGFP-Amph I FL, EGFP-Amph I (1-278), or EGFP-Amph I (279-695). Top: EGFP expression; middle, images of FM 4–64; bottom, overlay of EGFP expression and FM 4–64 labeling. FM 4–64 labeling in boutons was inhibited by overexpressing EGFP-Amph I (1-278) and EGFP-Amph I (279-695). The arrows indicate the FM 4–64 labeling in boutons. Scale bar, 20 μm. (**f**) DiI staining of dendritic spines. The density of dendritic spines in neurons expressing EGFP-Amph I (1-278) is lower than that in neurons expressing EGFP, EGFP-Amph I FL, or EGFP-Amph I (279-695). Scale bar, 20 μm. (**g, h**) Quantification of the transferrin uptake in COS-7 cells transfected with EGFP, EGFP-Amph I FL, EGFP-Amph I (1-278), EGFP-Amph I (279-695) (**g**); or EGFP and EGFP-active AEP (26-323) (**h**). The uptake of transferrin was calculated as the mean fluorescence density of 70–80 cells under each condition. The fluorescence density was normalized to the untransfected cells (mean ± s.e.m. n = 67–83; *p<0.05, **p<0. 01, ***p<0.001). (**i**) Quantitative analysis of FM 4–64 uptake. FM 4–64 labeling was calculated as the integral fluorescence intensity of 52–70 boutons under each condition. The mean values (± s.e.m) were expressed as the percentage of FM 4–64 labeling in untransfected boutons. ***p<0.001. (**j**) The dendritic spines density was quantified by calculating the number of dendritic branch per 100 μm dendrite. Data represent mean ± s.e.m. of five independent experiments. **p <0.01.

The online version of this article includes the following source data for figure 3:

**Source data 1.** Quantification of the transferrin uptake in COS-7 cells transfected with EGFP, EGFP-Amph I FL, EGFP-Amph I (1-278), and EGFP-Amph I (279-695).

**Source data 2.** Quantification of the transferrin uptake in COS-7 cells transfected with EGFP and EGFP- active AEP (26-323).

**Source data 3.** Quantitative analysis of FM 4-64 uptake.

**Source data 4.** Quantification of the dendritic spines density.

---

(1-278). As expected, AT8 and AT100 immunoreactivity were significantly decreased in the presence of roscovitine (*Figure 5f–h*).

Interestingly, we observed increased p35 immunoreactivity in boutons overexpressing Amphiphysin I (1-278), suggesting that p35/p25 may mediate the effect of Amphiphysin I (1-278) on promoting CDK5 activity (*Figure 5i*). To clarify the relationship between Amphiphysin I (1-278), p35, and tau phosphorylation, we co-transfected HEK293 cells with His-p35, EGFP-Tau, and GST-Amphiphysin I fragments, and then performed His pull-down assay. We found that Amphiphysin I (1-278) fragment bound to p35, and enhanced the interaction between p35 and tau (*Figure 5j*). When GST-Amphiphysin I fragments and Flag-His-p35 were co-transfected into HEK293 cells, Amphiphysin I (1-278) fragment was more abundant in the cell membrane than full-length Amphiphysin I, while Amphiphysin I (279-695) fragment mainly distributed in the cytoplasm. Furthermore, the expression of p25 was higher in the cytoplasm fraction in the presence of Amphiphysin I (1-278) (*Figure 5k*). Thus, Amphiphysin I (1-278) may sequester p35 in the membrane, where it is processed to p25 by calpain (*Minegishi et al., 2010*). These results suggest that Amphiphysin I (1-278) interacts with p35, activates CDK5, and induces the hyperphosphorylation of tau.

## Amphiphysin I fragments induce synaptic dysfunction and cognitive impairments in Tau P301S transgenic mice

We further investigated the effect of AEP-mediated Amphiphysin I fragmentation on disease progression in Tau P301S transgenic mice. The Tau P301S mice express P301S mutant human tau and develop widespread neurofibrillary tangle-like inclusions in the brain, which is accompanied by synaptic dysfunction and behavioral impairments (*Yoshiyama et al., 2007*). We injected AAVs encoding either EGFP, full-length Amphiphysin I, or AEP-derived Amphiphysin I fragments (1–278 and 279–695) into the hippocampal CA1 area of two-month-old Tau P301S mice. Two months later, we observed a strong expression of EGFP-tagged proteins in all mice (*Figure 6a*). Electron microscope analysis of brain sections showed that the density of synapse in the hippocampus of mice injected with AAV-EGFP-Amphiphysin I fragments (1–278 and 279–695) was reduced compared to the mice overexpressing full-length Amphiphysin I (*Figure 6b,c*). Golgi staining revealed that AEP-derived Amphiphysin I fragments (1–278 and 279–695) promoted the loss of dendritic spines in Tau P301S transgenic mice (*Figure 6d,e*).

Next, we assessed the effect of Amphiphysin I fragments on memory functions of Tau P301S transgenic mice in the Morris water maze test. During the training phase, the latency to find the platform was progressively decreased in the Tau P301S transgenic mice expressing EGFP,

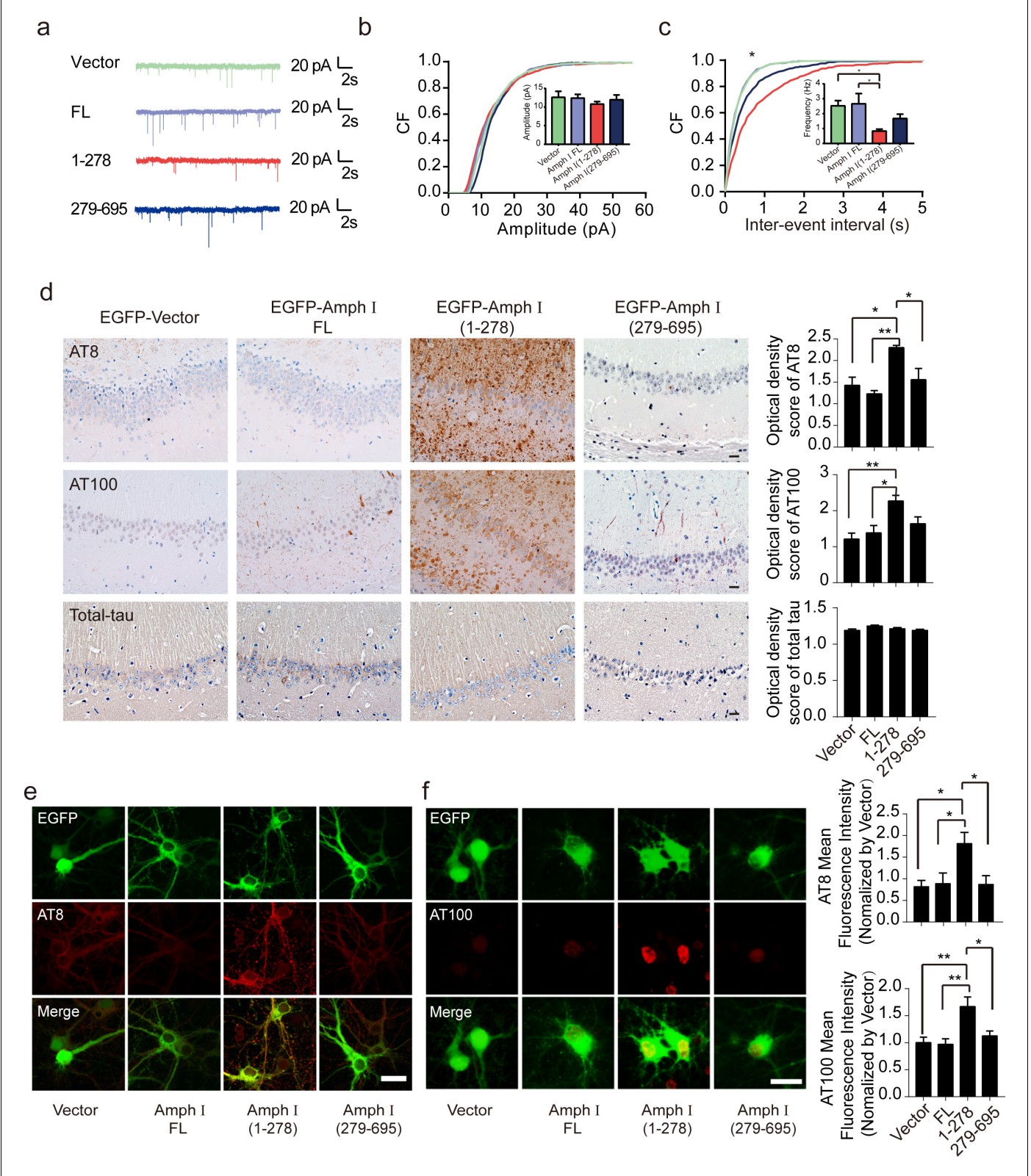

**Figure 4.** Amph I (1-278) fragment disrupts presynaptic function and promotes the hyperphosphorylation of tau. (**a**) Representative traces of miniature excitatory autaptic currents (mEACs) recorded from isolated hippocampal neurons expressing EGFP-Amph I FL, EGFP-Amph I (1-278), or EGFP-Amph I (279-695). (**b**) Cumulative plots and mean values (inset) of mEAC amplitude in isolated hippocampal neurons expressing full-length Amph I or its fragments. (**c**) Cumulative plots and mean values (inset) of mEAC frequency in isolated hippocampal neurons in various conditions. Kolmogorov-

*Figure 4 continued on next page*

*Figure 4 continued*

Smirnov test in c; one-way ANOVA with post hoc Dunnett's test in d; *p<0.05; **p<0.01; ***p<0.001; n = 5 independent experiment per group. Data are mean ± s.e.m. (**d**) AT8, AT100, and total-tau immunostaining of hippocampus in Tau P301S injected with adeno-associated virus (AAV) encoding Amph I FL, Amph I (1-278), or Amph I (279-695). Scale bar, 20 μm. Quantification of AT8, AT100, and total tau immunostaining was analyzed by Image J (mean ± s.e.m.; one-way ANOVA, n = 4 mice per group; *p<0.05, **p<0.01). (**e**) AT8 immunostaining of wild-type rat primary cortical neurons transfected with AAVs encoding EGFP-vector, EGFP-Amph I FL, EGFP-Amph I (1-278), and EGFP-Amph I (279-695). Scale bar, 40 μm. Quantification of AT8 immunostaining was analyzed by Image J. The fluorescence intensity was normalized to EGFP-vector group (mean ± s.e.m.; one-way ANOVA, n = 5 independent experiment per group; *p<0.05). (**f**) AT100 immunostaining of wild-type rat primary cortical neurons transfected with EGFP-vector, EGFP-Amph I FL, EGFP-Amph I (1-278), EGFP-Amph I (279-695). Scale bar, 20 μm. Quantification of AT100 immunostaining was analyzed by Image J. The fluorescence intensity was normalized to EGFP-vector group (mean ± s.e.m.; one-way ANOVA, n = 5 independent experiment per group; *p<0.05, **p<0.01).

The online version of this article includes the following source data and figure supplement(s) for figure 4:

**Source data 1.** Cumulative plots and mean values of mEAC amplitude in isolated hippocampal neurons expressing full-length Amph I or its fragments.

**Source data 2.** Cumulative plots and mean values of mEAC frequency in isolated hippocampal neurons expressing full-length Amph I or its fragments.

**Source data 3.** AT8, AT100 and total-tau immunostaining of hippocampus in Tau P301S injected with AAVs encoding Amph I FL, Amph I (1-278), or Amph I (279-695).

**Source data 4.** Quantification of AT8 and AT100 immunostaining were analyzed by Image J.

**Figure supplement 1.** Verification of the AT100 antibody.

demonstrating a learning effect. However, the learning ability was substantially impaired in mice expressing Amphiphysin I (1-278) (*Figure 6f,g*). In the probe trial, the mice expressing Amphiphysin I (1-278) showed decreased time in the target quadrant, suggesting impaired memory function (*Figure 6h*). All the mice showed comparable swim speeds (*Figure 6i*), suggesting that the overexpression of Amphiphysin I and its fragments does not affect motor function. Consistent with the water maze test results, the mice expressing Amphiphysin I (1-278) fragments spent less time in the new arm in the Y-maze test (*Figure 6j*). These results indicate that the Amphiphysin I (1-278) fragment induces learning and memory impairments. The long-term potentiation (LTP) of fEPSPs in the hippocampus is believed to be the basis of learning and memory (*Fedulov et al., 2007*; *Nicoll, 2017*). We found that LTP was diminished in mice overexpressing Amphiphysin I (1-278) and (279-596) fragments (*Figure 6k*). The averaged slope of fEPSPs was smaller in mice expressing Amphiphysin I (1-278) and (279-596) fragments than in those expressing EGFP or full-length Amphiphysin I (*Figure 6l*). Hence, the Amphiphysin I (1-278) and (279-596) fragments induce synaptic dysfunction in transgenic mouse models of AD.

## Overexpression of AEP-uncleavable mutant Amphiphysin I reverses synaptic dysfunction and cognitive impairments

To confirm the effects of AEP-generated Amphiphysin I fragments in AD pathology, we injected AAVs encoding wild-type Amphiphysin I or N278A mutant Amphiphysin I, which cannot be cleaved by AEP into the hippocampus of two-month-old Tau P301S transgenic mice. Two months later, we detected similar levels of EGFP-tagged wild-type and mutant Amphiphysin I in the hippocampus (*Figure 7a*). The density of hippocampal synapse in mice expressing AEP-uncleavable Amphiphysin I N278A mutant was much higher than that in mice expressing wild-type Amphiphysin I (*Figure 7b,c*). Overexpression of uncleavable Amphiphysin I (N278A) decreased AT8 immunoreactivity in hippocampus sections, compared with wild-type Amphiphysin I (*Figure 7d*). The water maze test found that mice expression Amphiphysin I N278A spent more time in the target quadrant in the probe test, indicating reversed memory function (*Figure 7e*). All groups of mice displayed comparable swim speeds (*Figure 7—figure supplement 1*), suggesting that the overexpression of wild-type and mutant Amphiphysin I does not affect motor function. The electrophysiological analysis found that the LTP was improved in mice injected with AAV-Amphiphysin I N278A compared with mice injected with AAV-wild-type Amphiphysin I (*Figure 7f,g*). These results indicate that the N278A mutation which blocks the fragmentation of Amphiphysin1 reverses synaptic dysfunction and cognitive impairments in Tau P301S mice.

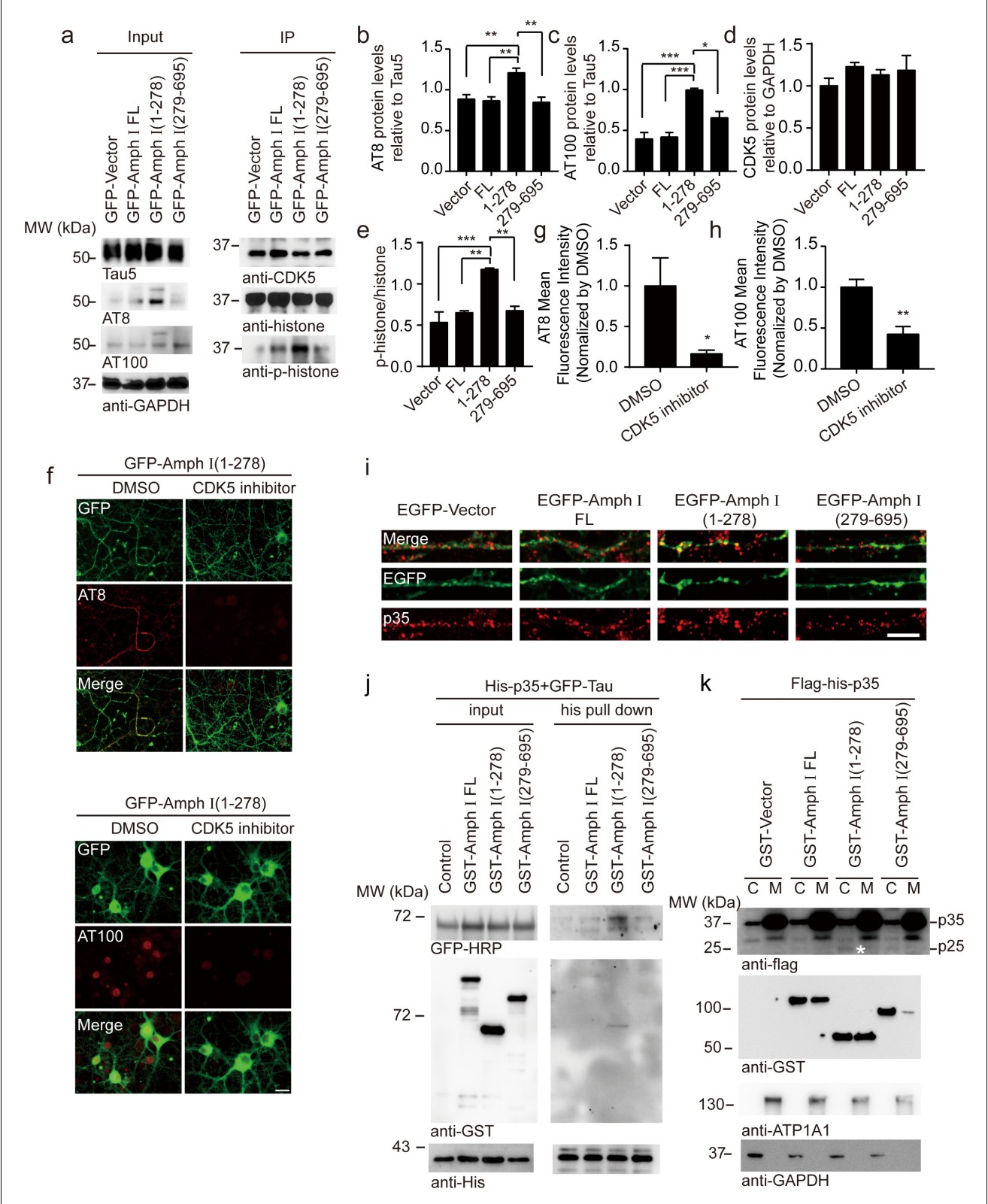

**Figure 5.** Amph I (1-278) fragment enhances CDK5 kinase activity. (**a**) Western blot showing the tau hyperphosphorylation of primary cortical neurons transfected with EGFP-vector, EGFP-Amph I FL, EGFP-Amph I (1-278), EGFP-Amph I (279-695). Left: The neuron lysate was detected by AT8, AT100, Tau5, and GAPDH antibodies. Right: The neuron lysate was immunoprecipitated with the anti-CDK5 antibody conjugated to protein A/G sepharose. The CDK5 activity was measured as the level of histone H1 phosphorylation. (**b–e**) Quantification of the immunoreactivity in (**a**) (mean ± s.e.m.; one-way

*Figure 5 continued on next page*

*Figure 5 continued*

ANOVA, n = 4 independent experiments per group; *p<0.05, **p<0.01, ***p<0.001). (f) AT8 and AT100 immunostaining of primary neurons showing the effect of 10 μM CDK5 inhibitor roscovitine on tau hyperphosphorylation. Scale bar, 20 μm. (g, h) Quantification of AT8, AT100 immunostaining shown in (f). The fluorescence intensity was normalized to dimethyl sulfoxide (DMSO) group (mean ± s.e.m.; one-way ANOVA, n = 5 independent experiment per group; *p<0.05, **p<0.01). (i) Immunostaining of primary neurons showing the localization of p35 in boutons expressing Amph I and its fragments. Scale bar, 10 μm. (j) His pull-down assay showing the interaction between p35, Tau, and Amph I fragments. His-p35, EGFP-Tau, GST-vector, GST-Amph I FL, GST-Amph I (1-278), or GST-Amph I (279-695) was expressed in HEK293 as indicated. (k) Western blot showing the distribution of Flag-His-p35 and GST-vector, GST-Amph I FL, GST-Amph I (1-278) or GST-Amph I (279-695) in cytoplasm and membrane. ATP1A1 is the marker of the membrane. Asterisks indicate p25. C, cytoplasm; M, membrane.

The online version of this article includes the following source data for figure 5:

**Source data 1.** Quantification of the immunoreactivity of AT8, AT100, CDK5, p-histone/histone.
**Source data 2.** Quantification of AT8, AT100 immunostaining .

## Discussion

In the present study, we identified a synaptic protein Amphiphysin I, which is cleaved by AEP in AD. AEP-generated Amphiphysin I fragments are found in the AD mouse model and human AD brain. The Amphiphysin I fragments disrupt clathrin-mediated endocytosis and induce synaptic dysfunction. Furthermore, Amphiphysin I (1-278) fragment induces tau hyperphosphorylation through activating CDK5. Therefore, our results indicate that Amphiphysin I fragments generated by AEP induce synaptic dysfunction and tau hyperphosphorylation, which are two of the major pathological features in the AD brain.

Amphiphysin I plays a pivotal role in clathrin-mediated endocytosis by interacting with dephosphins such as clathrin, dynamin, and Amphiphysin II (*David et al., 1996*; *McMahon et al., 1997*; *Wigge et al., 1997*). We found that Amphiphysin I (1-278) fragment loses the function of binding to clathrin and dynamin, whereas the C-terminal (279-695) fragment loses the function of binding to Amphiphysin II and the cell membrane (*Figures 3* and *5e*). During clathrin-mediated endocytosis, clathrin induces the deformation of the membrane into a budding vesicle (*Robinson, 1994*), whereas dynamin mediates the fission of the vesicle neck (*Hinshaw and Schmid, 1995*). Once Amphiphysin I is cleaved, the resultant N-terminal fragment binds to the membrane and competes with full-length Amphiphysin I, which may interrupt the recruiting of clathrin and dynamin from cytosol to the endocytic site, resulting in disrupted clathrin-mediated endocytosis. Besides, the C-terminal fragment of Amphiphysin I (279-695) recruits dynamin and clathrin but cannot help them translocate to the membrane. This was proven by the inhibition of transferrin uptake and FM 4–64 dye assay in COS-7 cells or neurons expressing Amphiphysin I fragments. Additionally, overexpression of Amphiphysin I fragments in mice induces synaptic loss, synaptic dysfunction, and memory deficits (*Figure 6*).

NFTs, which are composed of hyperphosphorylated tau, are a major characteristic of AD. Here, we found significant tau hyperphosphorylation in mice overexpressing Amphiphysin I (1-278) (*Figure 4*). Considering the interaction between Amphiphysin I and CDK5 activator p35 (*Floyd et al., 2001*), we assumed that Amphiphysin I (1-278) may activate CDK5 through promoting the cleavage of p35 by calpain. Indeed, we found Amphiphysin I (1-278) binds to p35 with higher affinity than full-length Amphiphysin I. Moreover, Amphiphysin I (1-278) is mainly localized in cell membranes, where p35 tends to be cleaved by calpain (*Minegishi et al., 2010*). As expected, more p35 is cleaved into p25 in the presence of Amphiphysin I (1-278). We also found increased CDK5 activity in neurons overexpression Amphiphysin I (1-278). Furthermore, inhibition of CDK5 reverses the phosphorylation of tau. Taken together, both the in vivo and in vitro evidence indicate that Amphiphysin I (1-278) promotes the hyperphosphorylation of tau by activating CDK5.

Here, we used the Tau P301S mice to explore the role of Amphiphysin I fragments on tau phosphorylation and synaptic dysfunction. The Tau P301S mice overexpress a P301S mutant human tau that is indicated in frontotemporal dementia (FTD). Since the mice develop tau pathology, synaptic dysfunction, and behavioral impairments (*Yoshiyama et al., 2007*), it is widely used for the investigation of tauopathies. Thus, the results in the present study may reflect a common mechanism in tauopathies. Synaptic dysfunction and tau hyperphosphorylation are key pathological characteristics of AD. As discussed above, the Amphiphysin I (1-278) fragment may induce synaptic dysfunction and tau phosphorylation through independent pathways. It causes synaptic dysfunction by interfering

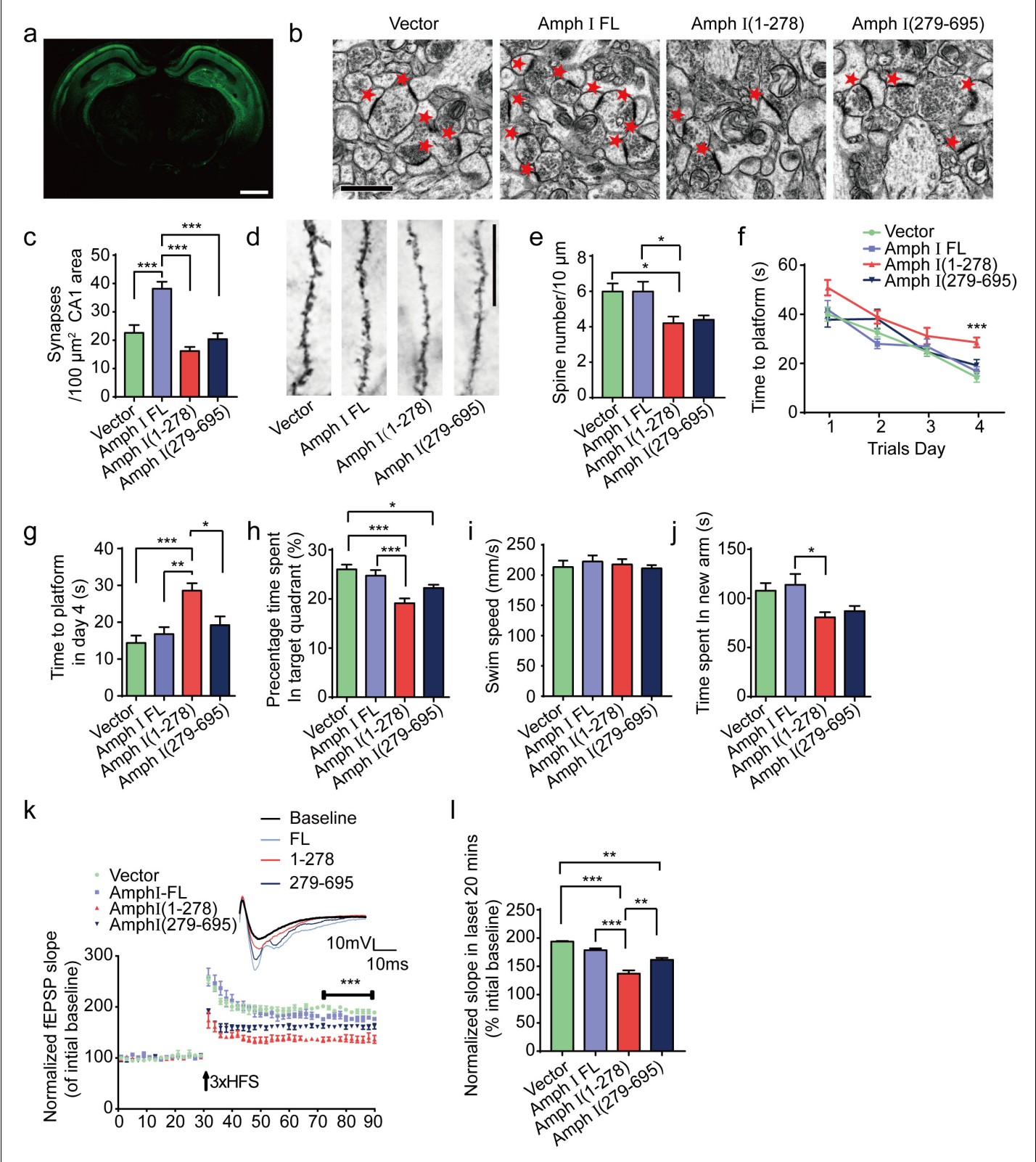

**Figure 6.** Amph I fragments induce synaptic dysfunction and cognitive impairments in Tau P301S transgenic mice. (**a**) The expression of EGFP in Tau P301S mice injected with AAV-EGFP. Scale bar, 1 mm. (**b**) Electron microscopy of synapses. Arrows indicate the synapses. Scale bar, 5 μm. (**c**) Quantification of synaptic density (mean ± s.e.m.; n = 5 mice per group; ***p<0.001, one-way ANOVA). (**d**) Golgi staining revealed the dendritic spines from the apical dendritic layer of the CA1 region. Scale bar, 20 μm. (**e**) Quantification of spine density (mean ± s.e.m.; n = 5 mice per group; *p<0.05,

*Figure 6 continued on next page*

*Figure 6 continued*

one-way ANOVA). (f, g) Morris water maze analysis as the escape latency (g) and escape latency in day 4 (h) (mean ± s.e.m.; n = 8; ***p<0.001, one-way ANOVA). (h) Probe trial of Morris water maze test (mean ± s.e.m.; n = 8 mice per group; *p<0.05, **p<0.01, one-way ANOVA). (i) Swim speed of mice injected with AAVs encoding EGFP, EGFP-Amph I FL, EGFP-Amph I (1-278), and EGFP-Amph I (279-695) (mean ± s.e.m.; n = 8 mice per group; one-way ANOVA). (j) Y-maze analysis as time spent in new arms (mean ± s.e.m.; n = 8 mice per group; *p<0.05, **p<0.01, one-way ANOVA). (k) The slope of fEPSP after HFS recorded on hippocampal slices. Arrow indicates HFS onset. Shown traces are representative fEPSPs of three samples recorded before and after LTP induction. (l) Quantitative analyses for normalized fEPSPs 70–90 min after HFS (mean ± s.e.m.; n = 3 mice per group; ***p<0.001, two-way ANOVA). AAV, adeno-associated virus; HFS, high-frequency stimulation; LTP, long-term potentiation.

The online version of this article includes the following source data for figure 6:

**Source data 1.** Quantification of synaptic density.
**Source data 2.** Quantification of spine number.
**Source data 3.** Escape latency of Morris water maze analysis.
**Source data 4.** Escape latency of Morris water maze analysis in day 4.
**Source data 5.** Probe trial of Morris water maze test.
**Source data 6.** Swim speed of mice injected with AAVs encoding EGFP, EGFP-Amph I FL, EGFP-Amph I (1-278), and EGFP-Amph I (279-695).
**Source data 7.** Y maze analysis as time spent in new arms.
**Source data 8.** The slope of fEPSP after HFS recorded on hippocampal slices.
**Source data 9.** Quantitative analyses for the average of normalized fEPSPs 70-90 min after HFS.

with clathrin-mediated endocytosis and synaptic vesicle recycling and enhances tau hyperphosphorylation through the p35/CDK5 pathway. However, it has been shown that tau hyperphosphorylation induces synaptic dysfunction. Thus, the hyperphosphorylated tau induced by Amphiphysin I (1-278) may also contribute to synaptic dysfunction.

Amphiphysin I is an endocytosis-related protein that plays an important role in the reuptake of neurotransmitters and the formation of synaptic vesicles. The dysfunction of synaptic vesicle recycling is found in early AD and is believed to contribute to cognitive impairments (*Arendt, 2009*). The dysfunction of many endocytic proteins is involved in AD, such as Amphiphysin II, CD2AP, PICALM, RIN3, and so on *Karch and Goate, 2015*. Here, we show that Amphiphysin I fragments generated by AEP interrupt synaptic vesicle endocytosis and induce synaptic dysfunction, as indicated by impaired mEPSCs in cultured neurons and diminished LTP in brain slides expressing Amphiphysin I fragments. In the Tau P301S mouse, overexpression of the full-length Amphiphysin I exerted protective effects, which can be explained by the beneficial effect of Amphiphysin I on synaptic function. It is noteworthy that the overexpressed full-length Amphiphysin I could also be fragmented by endogenous AEP and generate toxic fragments. However, the protective effect of the full-length Amphiphysin I may exceed the toxic effect of the AEP-generated Amphiphysin I fragment in our experiments.

The activity of Amphiphysin I is also regulated by phosphorylation which inhibits its binding to AP-2 and clathrin (*Cousin and Robinson, 2001*; *Slepnev et al., 1998*). Furthermore, calpain has been reported to regulate the activity of Amphiphysin I. During neural hyperexcitation, calpain cleaves Amphiphysin I, thus inhibits synaptic vesicle recycling, which may play an auto-protective role against neural hyperexcitation (*Wu et al., 2007*). Here, we found a novel posttranslational regulation of Amphiphysin I, cleavage by AEP. Noticeably, AEP cleaves Amphiphysin I in an age-dependent manner. Overexpression of AEP-derived Amphiphysin I fragments in the AD mouse model induces synaptic dysfunction, tau hyperphosphorylation, and neurotoxicity. Nevertheless, overexpression of the Amphiphysin I mutation which is resistant to the cleavage by AEP partially reversed AD pathology. Overall, our results suggest that the AEP-mediated cleavage of Amphiphysin I may take part in the pathogenesis of AD. Blocking Amphiphysin I cleavage by AEP may be a novel therapeutic intervention for treating AD.

# Materials and methods

**Key resources table**

*Continued on next page*

*Continued*

| Reagent type (species) or resource | Designation | Source or reference | Identifiers | Additional information |
|---|---|---|---|---|
| Reagent type (species) or resource | Designation | Source or reference | Identifiers | Additional information |
| Antibody | Anti-GST | Proteintech | Cat # 66001–2-Ig | WB (1:5000) |
|  | (mouse monoclonal) |  | RRID:AB_2881488 |  |
| Antibody | EGFP-HRP | Proteintech | Cat # HRP-66002 | WB (1:5000) |
|  | (mouse monoclonal) |  | RRID:AB_2883834 |  |
| Antibody | Anti-HIS | Proteintech | Cat # 66005–1-Ig | WB (1:5000) |
|  | (mouse monoclonal) |  | RRID:AB_11232599 |  |
| Antibody | Anti-GAPDH | Proteintech | Cat # 60004–1-Ig | WB (1:5000) |
|  | (mouse monoclonal) |  | RRID:AB_2107436 |  |
| Antibody | Anti-Amphiphysin I | Santa cruz | Cat # sc-21710 | WB (1:1000) |
|  | (mouse monoclonal) |  | RRID:AB_673386 |  |
| Antibody | Anti-Amphiphysin 2 | Santa cruz | Cat # sc-23918 | WB (1:1000) |
|  | (mouse monoclonal) |  | RRID:AB_667901 |  |
| Antibody | Anti-dynamin 1 | Proteintech | Cat # 18205–1-AP | WB (1:1000) |
|  | (mouse monoclonal) |  | RRID:AB_2093197 |  |
| Antibody | AT8 | Thermo | Cat # MN1020 | WB/IF/IHC (1:1000) |
|  | (mouse monoclonal) |  | RRID:AB_223647 |  |
| Antibody | AT100 | Thermo | Cat # MN1060 | WB/IF/IHC (1:1000) |
|  | (mouse monoclonal) |  | RRID:AB_223652 |  |
| Antibody | tau5 | Thermo | Cat # MA5-12805 | WB/IHC (1:1000) |
|  | (mouse monoclonal) |  | RRID:AB_10978000 |  |
| Antibody | Anti-CDK5 | Santa cruz | Cat # sc-6247 | WB (1:1000) |
|  | (mouse monoclonal) |  | RRID:AB_627241 |  |
| Antibody | Anti-Histone H1 | Abcam | Cat # ab61177 | WB (1:1000) |
|  | (polyclonal antibody) |  | RRID:AB_941946 |  |
| Antibody | Anti-p-histone H1 | Abcam | Cat # ab3596 | WB (1:1000) |
|  | (polyclonal antibody) |  | RRID:AB_303939 |  |
| Chemical compound | Transferrin Alexa Fluor 546 | Thermo | Cat # T23364 |  |
| Chemical compound | FM 4–64 | Thermo | Cat # T13320 |  |
| Protein | Recombinant AEP | Sino Biological | Cat # 50051-M07H |  |

## Mice

Tau P301S mice on a C57BL/6J background (line PS19) and wild-type C57BL/6J mice were from The Jackson Laboratory (stock number: 008169 and 000664, respectively). Animal care and handling were performed according to the Declaration of Helsinki and the guidelines of Renmin Hospital, Wuhan University. The sample size was determined by Power and Precision (Biostat). Only male mice were used in this study. Animals were randomly assigned to different groups, and the investigators were blind to the group assignment during the animal experiments. The protocol was reviewed and approved by the Animal Care and Use Committee of Renmin Hospital of Wuhan University (20180811).

## Human tissue samples

Postmortem brain samples were dissected from the frozen brain of AD cases and age-matched control cases from the Emory Alzheimer's Disease Research Center. The diagnosis of AD was

neuropathologically confirmed. Informed consent was obtained from all subjects or their relatives. The study was approved by the Ethics Committee (WDRY2021-K028).

## Transfection and infection of the cells

HEK293 cells were from the American Type Culture Collection (ATCC) and tested for mycoplasma contamination before use. The HEK293 cells were transfected with plasmids encoding wild-type or point mutant mGST-Amphiphysin I, myc-AEP, myc-AEP C189S, pEGFP-N1-Vector, pEGFP-N1-Amphiphysin I, pEGFP-N1-Amphiphysin I (1-278), pEGFP-N1-Amphiphysin I (279-695), and Flag-His-p35 using polyethyleneimine (PEI). To overexpress Amphiphysin I and its fragments in neurons, we infected the neurons with adeno-associated virus (AAV) encoding GFP, Amphiphysin I-FL, Amphiphysin I (1-278), and Amphiphysin (279-695), respectively.

## In vitro Amphiphysin I cleavage assay and AEP activity assay

Amphiphysin I cleavage assay and AEP activity assay were performed as described previously (*Zhang et al., 2014*). Briefly, HEK293 cells were transfected with 5 µg mGST-Amphiphysin I plasmids. Cells were collected, washed, and lysed in AEP buffer (50 mM sodium citrate, 5 mM DTT, 0.1% CHAPS and pH 5.5, 0.5% Triton X-100) 48 hr after transfection, and centrifuged for 10 min at 14,000 g at 4 °C. Supernatants were incubated with recombinant AEP (5 µg ml$^{-1}$) at pH7.4 or pH6.0 at 37 °C for different time points. AENK was used to test the effect of AEP inhibitor on the cleavage of Amphiphysin I. To measure the cleavage of purified Amphiphysin I by AEP, GST-tagged Amphiphysin I was purified with glutathione beads and then incubated with recombinant AEP in AEP buffer. The samples were then boiled in a 1× SDS loading buffer and analyzed by immunoblotting. Tissue homogenates or cell lysates (10 µg) were incubated in 200 µl assay buffer (20 mM citric acid, 60 mM Na$_2$HPO$_4$, 1 mM EDTA, 0.1% CHAPS, and 1 mM DTT, pH 6.0) containing 20 µM AEP substrate Z-Ala-Ala-Asn-AMC (Bachem). AMC released by enzymatic cleavage was quantified by measuring at 460 nm in a fluorescence plate reader at 37 °C for 45 min in kinetic mode for 5 min.

## Western blot analysis

The mouse brain tissue or human tissue samples were lysed in lysis buffer (50 mM Tris, pH 7.4, 40 mM NaCl, 1 mM EDTA, 0.5% Triton X-100, 1.5 mM Na3VO4, 50 mM NaF, 10 mM sodium pyrophosphate and 10 mM sodium β-glycerophosphate, supplemented with protease inhibitors cocktail) and centrifuged for 15 min at 16,000 g. The supernatant was boiled in SDS loading buffer. After SDS-PAGE, the samples were transferred to the nitrocellulose membranes. The membranes were incubated with primary antibodies overnight at 4 °C. The membranes were washed three times in PBST and incubated with HRP-conjugated secondary antibodies. The signals were developed using enhanced chemiluminescent (ECL) substrates.

## Mass spectrometry analysis

AEP knockout mice and age-matched wild-type mice brain lysates samples were in-gel digested with trypsin. The samples were resuspended in loading buffer (0.1% formic acid, 0.03% trifluoroacetic acid, and 1% acetonitrile) and loaded onto a 20 cm nano-high performance liquid chromatography column (internal diameter 100 mm) packed with Reprosil-Pur 120 C18-AQ 1.9 mm beads (Dr. Maisch) and eluted over a 2 hr 4–80% buffer B reverse phase gradient (buffer A: 0.1% formic acid and 1% acetonitrile in water; buffer B: 0.1% formic acid in acetonitrile) generated by a NanoAcquity UPLC system (Waters Corporation). Samples were ionized on a hybrid LTQ XL Orbitrap mass spectrometer (Thermo) using a 2.0 kV electrospray ionization voltage from a nano-ESI source (Thermo). After collision-induced dissociation (collision energy 35%, activation Q 0.25, activation time 30 ms) for the top 10 precursor ions, data-dependent acquisition of centroid MS spectra at 30,000 resolution and MS/MS spectra were obtained in the LTQ, and the charge determined by the acquisition software is $z \geq 2$. Dynamic exclusion of peaks already sequenced was for 20 s with early expiration for two count events with signal-to-noise >2. Automatic gating control was set to 150 ms maximum injection time or 106 counts.

To identify AEP cleavage sites on Amphiphysin I, the SageNSorcerer SEQUEST 3.5 algorithm was used to search and match MS/MS spectra to a complete semi-tryptic mouse proteome database plus pseudo-reversed decoys sequences (*Elias and Gygi, 2007*; *Xu et al., 2009*) with a 20 p.p.m.

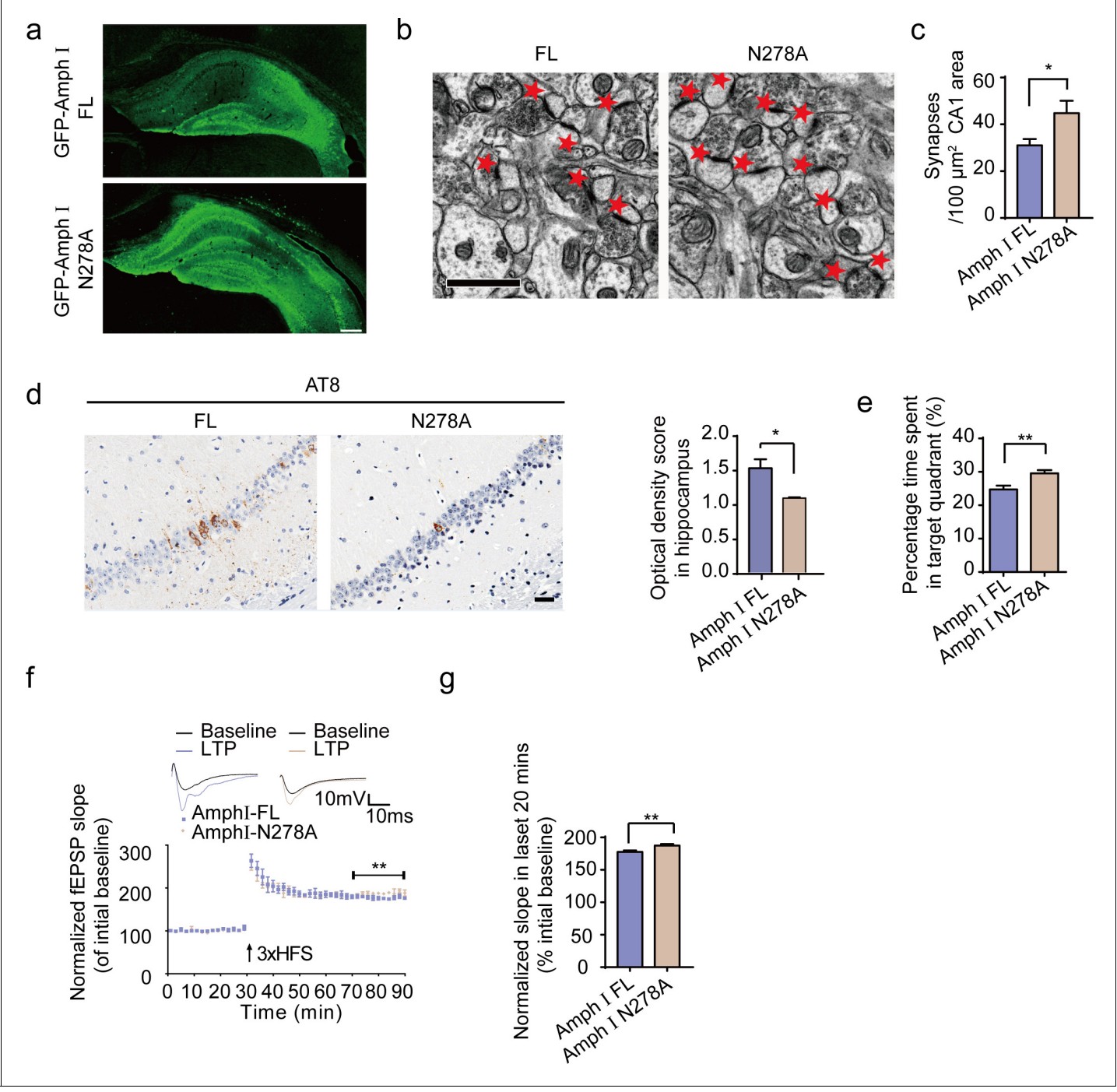

**Figure 7.** Expression of Amph1 N278A enhances synaptic and cognitive function in Tau P301S mice. (a) The expression of wild-type Amph I and N278A Amph I in Tau P301S mice. Scale bar, 200 μm. (b, c) Electron microscopy of synapses. Arrows indicate the synapses (n = 5 mice per group). Scale bar, 5 μm. (c) Quantification of synaptic density (mean ± s.e.m.; n = 5 mice per group; *p<0.05, Student's t-test). (d) AT8 immunostaining of hippocampus in Tau P301S injected with AAVs encoding Amph I FL and Amph I N278A. Scale bar, 20 μm. Quantification of the immunoreactivity (mean ± s.e.m.; one-way ANOVA, n = 4 mice per group; *p<0.05). (e) Probe trial of Morris water maze test (mean ± s.e.m.; n = 8 mice per group; *p<0.05, Student's t-test). (f) The slope of fEPSP after HFS recorded on hippocampal slices expressing wild-type and N278A mutant Amph I. Arrow indicates HFS onset. Shown traces are representative fEPSPs of three samples recorded before and after LTP induction. (g) Quantitative analyses for normalized fEPSPs 70–90 min after HFS (mean ± s.e.m.; n = 3 mice per group; **p<0.01, Paired t-test). AAV, adeno-associated virus; HFS, high-frequency stimulation; LTP, long-term potentiation.

The online version of this article includes the following source data and figure supplement(s) for figure 7:

**Source data 1.** Quantification of synaptic density.

*Figure 7 continued on next page*

*Figure 7 continued*

**Source data 2.** AT8 immunostaining of hippocampus in Tau P301S injected with AAVs encoding Amph I FL and Amph I N278A.
**Source data 3.** Probe trial of Morris water maze test.
**Source data 4.** The slope of fEPSP after HFS recorded on hippocampal slices.
**Source data 5.** Quantitative analyses for the normalized fEPSPs 70-90 min after HFS.
**Figure supplement 1.** Expression of Amph1 N278A does not affect motor function.
**Figure supplement 1—source data 1.** Swim speed of mice injected with AAVs encoding EGFP-Amph I FL and EGFP-Amph I N278A.

mass accuracy threshold. Only *b*- and *y*-ions were considered for scoring (Xcorr) and Xcorr along with DCn were dynamically increased for groups of peptides organized by a combination of trypticity (fully or partial) and precursor ion charge state to remove false-positive hits along with decoys until achieving a false-discovery rate (FDR) of <5% (<0.25% for proteins identified by more than one peptide). The FDR was estimated by the number of decoy matches (nd) and the total number of assigned matches (nt). FDR = 2*nd/nt, assuming mismatches in the original database were the same as in the decoy database. All semi-tryptic MS/MS spectra for putative AEP-generated Amphiphysin I cleavage sites were manually inspected.

## Transferrin uptake assay

COS-7 cells were transfected with pEGFP-N1 vector, pEGFP-N1-Amphiphysin I, pEGFP-N1-Amphiphysin I (1-278), and pEGFP-N1-Amphiphysin I (279-695), respectively. Two days after transfection, COS-7 cells were starved in serum-free DMEM for 30–60 min at 37 ℃. Then, cells were pulsed with 25 μg/ml Alexa Fluors 546-conjugated transferrin (Invitrogen) for 30 min at 37 ℃. After washed in PBS, the cells were fixed in 4% paraformaldehyde, and the signals were monitored under a confocal laser microscope (LEICA). To quantify the results, the fluorescence intensity of internalized transferrin was measured using ImageJ software (n = 70–80 cells). The fluorescence intensities of the transfected cells were normalized with non-transfected cells.

## FM dye uptake assay

FM dye imaging was performed as described previously (*Gaffield and Betz, 2006*). Briefly, to load the synaptic vesicles with the FM 4–64 dye, the neurons were incubated in high-K⁺ Tyrode's solution (90 mM) with 10 mM FM 4–64 dye for 1–2 min. The cells were switched to normal Tyrode's solution in the presence of FM 4–64 for 15–20 min to recover the nerve terminal. The staining solution was washed out of the imaging chamber with a dye-free solution. The FM 4–64 fluorescence intensity of boutons was measured by ImageJ software (n = 50–70 cells in each group).

## GST pull-down and His pull-down assay

For GST pull-down assay, bGST vector, bGST-Amphiphysin I-FL, bGST-Amphiphysin I (1-278), and bGST-Amphiphysin I (279-695) plasmids were transformed into competent BL21 cells, respectively. The expression was induced by adding 0.5 mM isopropyl-1-thio-β-D-galactopyranoside (IPTG). The cells were harvested and lysed in PBS and by sonication. The cell lysates were incubated with glutathione sepharose 4B GST-tagged protein purification resin. For His pull-down, Flag-His-p35 plasmids were transfected into HEK293 cells. The cells were lysed in lysis buffer and incubated with Ni-NTA Agarose. After washing four times with lysis buffer, the beads were incubated with mice brain tissue lysates or with lysates of HEK293 cells transiently transfected with indicated plasmids overnight. The beads were washed three times with PBST and boiled in SDS loading buffer for 10 min to elute the bound proteins.

## Immunostaining

Free-floating 30-μm-thick serial sections were treated with 0.3% H₂O₂ for 10 min and washed three times in PBS. Then, sections were blocked in 1% BSA, 0.3% Triton X-100, for 30 min and incubated with anti-Amphiphysin I N278 (generated and verified as described in the main text, 1:1000), AT8, or AT100 overnight at 4 ℃. The signal was developed using a Histostain-SP kit (Invitrogen). The images were analyzed by Image J, plus the IHC Profiler plugin. The program counts the pixels and evaluates the percentage contribution of high positive, positive, low positive, and negative. The 'optical

density score' was calculated as (percentage contribution of high positive*4 + percentage contribution of positive*3 + percentage contribution of low positive*2 + percentage contribution of negative*1)/100.

## AAV packaging and stereotaxic injection

The AAV particles encoding full-length Amphiphysin I, Amphiphysin I (1-278), Amphiphysin I (279-695) with the human synapsin I promoter were prepared by Shumi Technologies. Bilateral intracerebral injection of AAVs was performed stereotactically at coordinates posterior 2.5 mm, lateral 2.0 mm, ventral 1.7 mm relative to bregma in two-month-old Tau P301S mice. 250 nl of viral suspension containing $1 \times 10^9$ vector genomes (vg) was injected into each site using a 10 µl glass syringe with a fixed needle.

## Electron microscopy of synapses

Synaptic density was determined by electron microscopy as described previously (*Zhang et al., 2014*). After deep anesthesia, mice were perfused transcardially with 2% glutaraldehyde. Hippocampal slices were postfixed in cold 1% $OsO_4$ for 1 hr. Samples were prepared and examined using standard procedures. Ultrathin sections (90 nm) were stained with uranyl acetate and lead acetate and viewed at 100 kV in a JEOL 200CX electron microscope. Synapses were identified by the presence of synaptic vesicles and postsynaptic densities.

## Electrophysiology

The brain slides of Tau P301S mice injected with AAVs encoding full-length Amphiphysin I, Amphiphysin I (1-278), Amphiphysin I (279-695), and Amphiphysin I N278A were used in electrophysiology experiments. Mice were deeply anesthetized. When all pedal reflexes were abolished, brains were removed and dropped in an ice-cold oxygenated cutting solution containing the following: 25 mM D-glucose, 2.5 mM KCl, 1.26 mM $NaH_2PO_4$, 25 mM $NaHCO_3$, 7.2 mM $MgCl_2$, 0.5 mM $CaCl_2$, 3.1 mM Na-pyruvate, 11.35 mM ascorbic acid, and 97 mM choline chloride. Coronal slices (350 µm thick) containing the dorsal hippocampus were cut at 4℃ in the cutting solution using a Leica VT1000S vibratome and then transferred to an incubation chamber filled with oxygenated artificial cerebrospinal fluid (a-CSF), which contains the following: 118 mM NaCl, 2.5 mM KCl, 1 mM $NaH_2PO_4$, 26 mM $NaHCO_3$, 2 mM $MgCl_2$, 2 mM $CaCl_2$, and 22 mM glucose in a 35℃ water bath for 30 min and then put in room temperature for 30 min before being recorded. For LTP, brain slices were laid down in a chamber with an 8 × 8 microelectrode array in the bottom planar (each 50 × 50 µm in size, with an inter-polar distance of 150 µm) and kept submerged in a-CSF, with 1–2 mL/min continuing perfusion, with a platinum ring glued by nylon silk. When the noise was stable, electrophysiological signals were acquired using the MED64 System (Alpha MED Sciences, Panasonic). The fEPSPs in CA1 neurons were recorded by stimulating the Schaeffer fibers from CA3. LTP was induced by applying three trains of high-frequency stimulation (HFS; 100 Hz, 1 s duration).

For the electrophysiological recording of cultured neurons, the neurons cultured 14 d in vitro were transferred to a chamber perfused with the standard bath solution containing 145 mM NaCl, 5 mM KCl, 2 mM $CaCl_2$, 2 mM $MgCl_2$, 10 mM glucose and 10 HEPES (pH 7.40,~310 mOsm). Cultured neurons were recorded with patch pipettes (6–8 MΩ) filled with artificial intracellular fluid (100 mM $CsCH_3SO_3$, 20 mM KCl, 10 HEPES, 4 mM Mg-ATP, 0.3 mM Tris-GTP, 7 mM Tris2-Phosphocreatine, 3 mM QX-314, pH 7.3, 285–290 mOsm). Neurons were voltage-clamped at −70 mV with a Multiclamp 700B amplifier, and data were digitized with a Digidata 1550 and analyzed by pClamp 10.0 (Molecular Devices), mEACs were recorded at 32℃ in a bath solution containing 0.5 µM TTX and 10 µM bicuculline. Individual events were counted and analyzed with the MiniAnalysis program.

## Generation of antibody that specifically recognizes the AEP-generated Amphiphysin I fragment (anti-Amphiphysin I N278 antibody)

The anti-Amphiphysin I N278 antibody was generated by immunizing two rabbits with the peptide Ac-CLPSPTASPN. The rabbits were boosted four times with the immunizing peptides with 3-week intervals between injections. The titers against the immunizing peptide were determined by ELISA. The maximal dilution giving a positive response with the chromogenic substrate for horseradish

peroxidase was 1:512. The immunoactivity of the antiserum was further confirmed by Western blotting and immunohistochemistry.

## Morris water maze

Four-month-old Tau P301S mice were trained with extra maze cues as described previously (*Zhang et al., 2014*). Each subject was tested four times per day for four consecutive days with a 15 min intertrial interval. If the subjects did not find the platform within 60 s, they were manually guided to it. And the subjects were left on it for an additional 15 s after reaching the invisible platform. A probe trial was performed on day 5. The platform was removed and the percentage of time spent in the quadrant was measured over 60 s. All trials were analyzed for latency and swim speed using ANY-Maze software (San Diego Instruments).

## Y-maze test

The Y-maze test was carried out as described previously (*Ma et al., 2007*). Each arm is 40 cm long, 12 cm high, 3 cm wide at the bottom, and 10 cm wide at the top. The arms converge in an equilateral triangular central area that is 4 cm at its longest axis. The three arms were randomly designated. Start arm: the mouse started to explore, always open. Novel arm: blocked during the first trial but open during the second trial. The Y-maze test consisted of two trials separated by an inter-trial interval (ITI) to assess spatial recognition memory. The first trial (training) lasts for 5 min and allowed the mouse to explore only two arms (start arm and another arm) of the maze, with the third arm (novel arm) being blocked. The series of arm entries were recorded visually. After 2 hr ITI (*Wang et al., 2007*), the second trial (retention) was conducted, during which all three arms were accessible and novelty vs. familiarity was analyzed by comparing behavior in all three arms. For the second trial, the mouse was placed back in the maze in the same starting arm, with free access to all three arms for 5 min. Recordings were taken and later analyzed by ANY-Maze software, and the number of entries and time spent in each arm were analyzed by an investigator that is blind to the group assignment.

## Primary neuronal cultures

Primary rat cortical neurons dissected from E18 embryos were cultured as previously described (*Zhang et al., 2014*). AAV particles encoding full-length Amphiphysin I, Amphiphysin I (1-278), or Amphiphysin I (279-695) were added to the medium of neurons cultured 5 d in vitro. 1 week later, the neurons were fixed in 4% formaldehyde, permeabilized, and immunostained with AT8, AT100, TUNEL, and p35, respectively. The sections were covered with a glass cover using a mounting solution and examined under a fluorescence microscope (Olympus).

## CDK5 activity assay

As previously described (*Piedrahita et al., 2010*), the neuron expressing full-length Amphiphysin I or its fragments were homogenized in 1% NP-40 and then immunoprecipitated with anti-CDK5 antibody conjugated to protein A/G sepharose. The precipitated beads were incubated with purified histone H1(1 μg) in 50 μl kinase buffer (50 mM Tris-HCl, pH 7.2, 10 mM $MgCl_2$, and 1 mM dithiothreitol) for 30 min at 30 °C. The samples were boiled in an SDS loading buffer and were detected by immunoblotting with the antibody against histone H1, p-histone H1, CDK5, and p35, respectively.

## Statistical analyses

Statistical analysis was performed using either Student's t-test (two-group comparison) or one-way ANOVA (more than two groups). We first estimated the variation of the data and checked whether they meet normal distribution. If not stated otherwise, the variances were similar between the groups that are being compared. Differences with p-values <0.05 were considered significant.

## Acknowledgements

This work was supported by grants from the National Natural Science Foundation of China (No. 81822016, 81771382, and 81571249) to ZZ. We thank Dr. De Camilli P (Yale University School of Medicine) for the gift of the pGEX6-1-Amphiphysin-1 plasmid.

## Additional information

### Funding

| Funder | Grant reference number | Author |
|---|---|---|
| National Natural Science Foundation of China | 81822016 | Zhentao Zhang |
| National Natural Science Foundation of China | 81771382 | Zhentao Zhang |
| National Natural Science Foundation of China | 81571249 | Zhentao Zhang |

The funders had no role in study design, data collection and interpretation, or the decision to submit the work for publication.

### Author contributions

Xingyu Zhang, Investigation, Methodology, Writing - original draft; Li Zou, Investigation, Electrophysiology; Lanxia Meng, Min Xiong, Investigation, Methodology; Lina Pan, Li Tang, Investigation; Guiqin Chen, Yongfa Zheng, Jing Xiong, Methodology; Zhihao Wang, Zhaohui Zhang, Methodology, Writing - review and editing; Duc M Duong, Mass spectrometry; Xuebing Cao, Tao Wang, Keqiang Ye, Writing - review and editing; Zhentao Zhang, Funding acquisition, Investigation, Methodology, Project administration, Writing - review and editing

### Author ORCIDs

Zhentao Zhang (iD) https://orcid.org/0000-0001-6708-1472

### Ethics

Human subjects: The study was approved by the Ethics Committee (WDRY2021-K028). Informed consent, and consent to publish, was obtained.

Animal experimentation: Animal care and handling were performed according to the Declaration of Helsinki and guidelines of Renmin Hospital, Wuhan University. The protocol was reviewed and approved by the Animal Care and Use Committee of Renmin Hospital of Wuhan University (#20180811).

### Decision letter and Author response

Decision letter https://doi.org/10.7554/eLife.65301.sa1
Author response https://doi.org/10.7554/eLife.65301.sa2

## Additional files

### Supplementary files

• Supplementary file 1. The demographic and pathological characteristics of human samples.

• Transparent reporting form

### Data availability

All data generated or analysed during this study are included in the manuscript and supporting files.

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
