## [Decision Letter]

**Acceptance summary:**

This paper is of interest to a broad range of neuroscientists, especially in the field of Alzheimer's disease (AD) research. The study reveals a novel molecular mechanism that may contribute to the pathogenesis of tau pathology in AD and potentially in other tauopathies. By revealing a novel cleavage site in Amphiphysin I, that can be processed by the protease asparagine endopeptidase (AEP), these data provide a new perspective into the mechanisms that may underlie tau pathology, synaptic dysfunction, and memory deficits associated with neurodegenerative diseases.

**Decision letter after peer review:**

Thank you for submitting your article "Amphiphysin I cleavage by asparagine endopeptidase leads to tau hyperphosphorylation and synaptic dysfunction" for consideration by *eLife*. Your article has been reviewed by 3 peer reviewers, including Jeannie Chin as the Reviewing Editor and Reviewer #1, and the evaluation has been overseen by Gary Westbrook as the Senior Editor. The following individual involved in review of your submission have agreed to reveal their identity: Yadong Huang (Reviewer #2).

The reviewers have discussed the reviews with one another and the Reviewing Editor has drafted this decision to help you prepare a revised submission.

Summary:

This paper is of interest to a broad range of neuroscientists, especially in the field of Alzheimer's disease (AD) research. The study reveals a novel molecular mechanism that may contribute to the pathogenesis of tau pathology in AD and potentially in other tauopathies. The authors describe a novel cleavage site in Amphiphysin I, that can be processed by the protease asparagine endopeptidase (AEP). They demonstrate that the cleavage occurs at N278, and that the resulting fragments can lead to dysfunction in clathrin-mediated endocytosis, tau hyperphosphorylation, synaptic dysfunction, and impaired learning and memory. Overall, the data are compelling, although a few aspects of quantitative data analysis could be worked out better, and some of the narrative should be toned down to more accurately reflect what the data demonstrate.

Essential revisions:

1. The representative images of biochemical and IHC data are generally compelling. However, consistent inclusion of group data and quantification are necessary to support the conclusions of each experiment, particularly for Figure 2d, Figure 4d, 4e, Figure 5a, 5b, and Figure 7d.

2. In Figure 2, relevance of the Amph1 (1-278) fragment to aging/AD is provided by examination of brain samples of AD patients as well as a mouse model of tauopathy (P301S mice). Both western blots and immunohistochemistry are used, but not in parallel, across aging wildtype mice, P301S mice, and human patients. Western blot quantification of Amph1 (1-278) fragment in P301S mice and in AD brain samples, as well as co-immunostaining of anti-N278 with p-tau in AD brain sections, should be performed for better understanding the roles of AEP-cleaved Amphiphysin I (1-278) fragment in AD-related tau pathology. In addition, because the immunostaining of both mouse and human brain sections with anti- N278 is fairly diffuse, it is unclear how single positive cells were counted. Better images are needed to enable visualization of stained cells. Please also include demographic and pathological/clinical data of the patients from whom samples were analyzed.

3. Throughout the manuscript, the narrative needs to be toned down, particularly regarding the major claims, to more accurately reflect what the data demonstrate:

a. The functional experiments are performed in overexpression paradigms, and thus investigate the function of the respective Amphiphysin fragments directly. While this strategy allows investigation of the effects of the fragments on cellular function, it does not permit conclusions about the role of endogenous Amph1 cleavage. Therefore, statements such as "Amphiphysin 1 cleavage by AEP induces synaptic dysfunction" are not fully supported by the data. The authors would have had to demonstrate that blocking endogenous cleavage of Amph1 prevents synaptic dysfunction, which was not done. "Amphiphysin 1 (1-278) fragment induces synaptic dysfunction" would be more appropriate here.

b. Similarly, statements throughout the manuscript related to Figure 7 such as "Blockage of AmphI fragmentation by AEP ameliorates synaptic dysfunction and cognitive impairments in Tau P301S mice" are not fully supported by the data, since the authors did not block endogenous AEP-mediated cleavage of Amph1, but rather overexpressed a non-cleavable form in P301S mice. Endogenous Amph1 is likely still there and still cleaved. Moreover, since there were no wild-type mice to compare the P301S mice to, the authors did not demonstrate that P301S mice had synaptic dysfunction and cognitive impairments that were improved by expression of the non-cleavable Amph1.

c. Although the results overall implicate a role for Amphiphysin 1 cleavage in AD, it was not directly demonstrated in the current study. Therefore, statements such as "Cleavage of Amphiphysin 1 …. is required for the progression of AD…" (Line 301-302, but also throughout the manuscript) should be appropriately altered.

Toning down these and other similar statements throughout the manuscript will allow the conclusions to more accurately reflect the data.

4. In Figure 2d, in addition to the blot with the N278 antibody against cleaved Amph1, a blot for total Amph1 should be shown. This would clarify any potential change in total protein levels that may occur with aging, and furthermore show whether the AEP-generated fragment is indeed a major cleavage fragment or one of many. There are three bands visible on the N278 blot. What is the expected molecular weight for the 1-278 Amph1 fragment?

5. In Figure 4d, the increase in tau phosphorylation for mice expressing Amph1 (1-278) is striking. Do total tau levels change under these conditions?

6. In Figures4f and 5b, The AT100 staining shows an unusual, nuclear pattern – Tau is not usually detected in the nucleus. Why would this be the case? Do the authors see Tau in the nucleus in their system? Is the AT100 signal abolished in a Tau knockout/knockdown system?

7. In Figure 5e, the purity of the fractions should be demonstrated using marker proteins. Additionally, there is no p25 band visible on the blot, therefore the conclusion that more cytoplasmic p25 is observed in the presence of the 1-278 fragment is not supported.

8. In Figure 6c, synapse numbers are unaltered in mice expressing the two Amphiphysin fragments, they are however increased in mice overexpressing full-length Amphiphysin. The corresponding text on lines 256-258 needs to be changed to accurately reflect the data. Also, some discussion is warranted regarding whether the overexpressed full-length Amphiphysin gets cleaved by endogenous AEP, and why there are no obvious detrimental effects of the resulting fragments.

9. The authors showed that the Amph I (1-278) fragment could cause both tau hyperphosphorylation and synaptic dysfunctions in Tau P301S mice. It is not clear in terms of the relationship between these two phenotypes – does one lead to the other or are both in parallel downstream of Amph I fragment? The authors should at least discuss different possibilities in the Discussion.

10. Tau-P301S is associated with FTD, but not AD, although the Tau-P301S mouse model (PS19) is useful for tauopathy studies. The authors should discuss this clearly in the Discussion section.

[Editors' note: further revisions were suggested prior to acceptance, as described below.]

Thank you for submitting your article "Amphiphysin I cleavage by asparagine endopeptidase leads to tau hyperphosphorylation and synaptic dysfunction" for consideration by *eLife*. Your article has been reviewed by a Reviewing Editor, with Gary Westbrook as the Senior Editor. The Reviewing Editor has drafted these comments to help you prepare a revised submission.

The authors have substantially revised the manuscript, including new data, images, and analyses that strengthen the conclusions. A few comments remain.

Essential Revisions:

1. In the Methods sections, a description of how the immunohistochemical images were quantified is still lacking. Please define "Optical density score", and indicate whether this refers to areas of tissue or within individual cells.

2. In Figure 4D, AT8 and AT100 staining in the EGFP-Amph1 (279-695) group is markedly more intense than that of either EGFP-Vector or EGFP-Amph1 FL groups. However, the quantification graphs indicate that there is no difference between staining in these three groups. Why is this? Was the "Optical Density Score" calculated the same way as in Figure 2D and 2F?

3. In the legends for Figure 4 and 5, n's are listed as "n = 4" or n = 5". It seems that this should be corrected to "n = 4 mice per group" or "n = 5 mice per group", etc.

4. The authors have added a western blot of ATP1A1 to illustrate the purity of their membrane preparations, in Figure 5. However, there is no description of this new data in the text or in the figure legend.

5. The authors have toned down a number of their statements to accurately reflect the data. However, the title for the legend for Figure 7 still needs to be revised. It reads "Amph1 N278A mutation ameliorates synaptic dysfunction and cognitive deficits in Tau P301S mice. However, since no data from wildtype mice is included in this figure, it is not clear that the P301S mice exhibited synaptic dysfunction and cognitive deficits. It would be more accurate to state something like "Expression of Amph1 N278A enhances synaptic and cognitive function in Tau P301S mice."

6. A few typos still remain: Figure 6F X-axis label should read "Trials", not "Trails"; in Line 167, "loss" should be "lose". There are a number of minor grammatical errors throughout the text that could be easily corrected by a proof-reader.

[Editors' note: further revisions were suggested prior to acceptance, as described below.]

Thank you for resubmitting your work entitled "Amphiphysin I cleavage by asparagine endopeptidase leads to tau hyperphosphorylation and synaptic dysfunction" for further consideration by *eLife*. Your revised article has been evaluated by Gary Westbrook (Senior Editor) and a Reviewing Editor. The manuscript has been improved but there is one remaining issue that needs to be addressed, as described below (original comment and your response copied here for clarity):

Comment 2. In Figure 4D, AT8 and AT100 staining in the EGFP-Amph1 (279-695) group is markedly more intense than that of either EGFP-Vector or EGFP-Amph1 FL groups. However, the quantification graphs indicate that there is no difference between staining in these three groups. Why is this? Was the "Optical Density Score" calculated the same way as in Figure 2D and 2F?"

Your response: "The "Optical Density Score" calculated the same way as in Figure 2D and 2F. However, we calculated the optical density score of the entire image. The positive area of EGFP-Amph1 (279-695) group only accounted for a small percentage of the total area, which may explain why the difference between these groups were small."

The images shown in Figure 4D show alterations in expression that are just as striking, or even more so, than those shown in Figure 2D and 2F. Yet, the accompanying graphs indicate that the expression levels between the groups in question in 4D are not different, whereas those in 2D and 2F are markedly and significantly different. There is an incongruency that needs to be resolved. This brings up important questions of how representative the images are, whether similar sized and similarly magnified images were quantified, and how robust your system of image selection and quantification is. Please either perform the quantification at the same magnifications, or provide images in Figure 4D that are more representative of the quantification results.

---

## [Author Response]

Essential revisions:1. The representative images of biochemical and IHC data are generally compelling. However, consistent inclusion of group data and quantification are necessary to support the conclusions of each experiment, particularly for Figure 2d, Figure 4d, 4e, Figure 5a, 5b, and Figure 7d.

We appreciate the reviewer’s comments. The group data and quantification have been provided in the revised figures.

2. In Figure 2, relevance of the Amph1 (1-278) fragment to aging/AD is provided by examination of brain samples of AD patients as well as a mouse model of tauopathy (P301S mice). Both western blots and immunohistochemistry are used, but not in parallel, across aging wildtype mice, P301S mice, and human patients. Western blot quantification of Amph1 (1-278) fragment in P301S mice and in AD brain samples, as well as co-immunostaining of anti-N278 with p-tau in AD brain sections, should be performed for better understanding the roles of AEP-cleaved Amphiphysin I (1-278) fragment in AD-related tau pathology. In addition, because the immunostaining of both mouse and human brain sections with anti- N278 is fairly diffuse, it is unclear how single positive cells were counted. Better images are needed to enable visualization of stained cells. Please also include demographic and pathological/clinical data of the patients from whom samples were analyzed.

As suggested by the reviewer, we provided the co-immunostaining of anti-N278 with p-tau in AD brain sections in revised figure 2e. Western blot quantification of Amph1 (1-278) fragment in wild-type mouse, Tau P301S mice, AD and control human brain lysates have been shown in Supplementary Figure 2a-f. More clear images of immunohistochemistry are provided in Figure 2f to clearly show the stains. The quantification of IHC was performed by calculating the IHC optical density score using Image J (Revised Figures 2d, 2f, 4d, 7d). The demographic and pathological/clinical data of the patients are provided in Supplementary File 1.

3. Throughout the manuscript, the narrative needs to be toned down, particularly regarding the major claims, to more accurately reflect what the data demonstrate:a. The functional experiments are performed in overexpression paradigms, and thus investigate the function of the respective Amphiphysin fragments directly. While this strategy allows investigation of the effects of the fragments on cellular function, it does not permit conclusions about the role of endogenous Amph1 cleavage. Therefore, statements such as "Amphiphysin 1 cleavage by AEP induces synaptic dysfunction" are not fully supported by the data. The authors would have had to demonstrate that blocking endogenous cleavage of Amph1 prevents synaptic dysfunction, which was not done. "Amphiphysin 1 (1-278) fragment induces synaptic dysfunction" would be more appropriate here.b. Similarly, statements throughout the manuscript related to Figure 7 such as "Blockage of AmphI fragmentation by AEP ameliorates synaptic dysfunction and cognitive impairments in Tau P301S mice" are not fully supported by the data, since the authors did not block endogenous AEP-mediated cleavage of Amph1, but rather overexpressed a non-cleavable form in P301S mice. Endogenous Amph1 is likely still there and still cleaved. Moreover, since there were no wild-type mice to compare the P301S mice to, the authors did not demonstrate that P301S mice had synaptic dysfunction and cognitive impairments that were improved by expression of the non-cleavable Amph1.c. Although the results overall implicate a role for Amphiphysin 1 cleavage in AD, it was not directly demonstrated in the current study. Therefore, statements such as "Cleavage of Amphiphysin 1 …. is required for the progression of AD…" (Line 301-302, but also throughout the manuscript) should be appropriately altered.Toning down these and other similar statements throughout the manuscript will allow the conclusions to more accurately reflect the data.

We appreciate the reviewer’s comments. We have thoroughly revised the manuscript to properly describe our findings. The changes are highlighted in the revised manuscript.

4. In Figure 2d, in addition to the blot with the N278 antibody against cleaved Amph1, a blot for total Amph1 should be shown. This would clarify any potential change in total protein levels that may occur with aging, and furthermore show whether the AEP-generated fragment is indeed a major cleavage fragment or one of many. There are three bands visible on the N278 blot. What is the expected molecular weight for the 1-278 Amph1 fragment?

As suggested by the reviewer, we provided the blot for total Amph1 in Figure 2—figure supplement 2a,c. To figure out which band is the AMPH1(1-278) fragment, we used brain lysates from AEP knockout mice as the negative control, and wild-type brain lysates as the positive control. We found that the band with a molecular weight of about 55 kDa is the Amph1(1-278) fragment.

5. In Figure 4d, the increase in tau phosphorylation for mice expressing Amph1 (1-278) is striking. Do total tau levels change under these conditions?

To investigate the levels of total tau, we stained the slides with tau5 antibody, which recognizes the total tau. We found that the total tau levels were similar in different groups (Figure 4d, lower panel). These results indicate that the AMPH1(1-278) fragment enhances the phosphorylation of tau, but does not increase the levels of total tau. We discussed this issue in the revised manuscript (Page 10, lines 209-212).

6. In Figures 4f and 5b, The AT100 staining shows an unusual, nuclear pattern – Tau is not usually detected in the nucleus. Why would this be the case? Do the authors see Tau in the nucleus in their system? Is the AT100 signal abolished in a Tau knockout/knockdown system?

We noticed the nuclear localization of the AT100 signals in our experiments. The AT100 antibody recognizes p-tau phosphorylated at Thr212/Ser214 residues, which has been reported to be located in the nucleus of human brain samples (Hernandez-Ortega, Brain Pathol, 2015, 26: 593-605) and mouse neurons (Gartner, Neurobiol Aging, 1998, 19: 535-543). Actually, it has been demonstrated that AT100-positive tau is localized in the nucleolus of pyramidal cells from the CA1 region, and the AT100-positive nuclear tau in the cell nuclei increases progressively during aging (Gil L, et al., Brain Res, 2017, 1677:129-137). We also reported the nuclear localization of AT100 signals in Tau P301S mice (Zhang Z, et al., Nat Commun, 2017, 8:14740). To confirm the specificity of this antibody, we performed immunofluorescence staining with AT100 antibody in Tau knockout mice brain, and found that the signals were completely abolished (Figure 4—figure supplement 1). We discussed this issue in the revised manuscript (Page 10, Line 214-219).

7. In Figure 5e, the purity of the fractions should be demonstrated using marker proteins. Additionally, there is no p25 band visible on the blot, therefore the conclusion that more cytoplasmic p25 is observed in the presence of the 1-278 fragment is not supported.

As suggested by the reviewer, we confirmed the purity of the fractions using a plasma membrane marker ATP1A1. We provided a longer exposure Western blot to show the presence of p25 in revised Figure 5k (marked with an asterisk).

8. In Figure 6c, synapse numbers are unaltered in mice expressing the two Amphiphysin fragments, they are however increased in mice overexpressing full-length Amphiphysin. The corresponding text on lines 256-258 needs to be changed to accurately reflect the data. Also, some discussion is warranted regarding whether the overexpressed full-length Amphiphysin gets cleaved by endogenous AEP, and why there are no obvious detrimental effects of the resulting fragments.

As suggested by the reviewer, we revised the text to more accurately describe the results (Page 13, lines 269-272). We also discussed the issue that overexpressed Amphiphysin I can be cleaved by endogenous AEP, but no obvious detrimental effects were found (Page 18, lines 383-389). In the Tau P301S mouse, overexpression of the full-length Amphiphysin I exerted protective effects, which can be explained by the beneficial effect of Amphiphysin I on synaptic function. It is noteworthy that the overexpressed full-length Amphiphysin I could also be fragmented by endogenous AEP and generate toxic fragments. However, the protective effect of the full-length Amphiphysin I may exceed the toxic effect of the AEP-generated Amphiphysin I fragment in our experiments.

9. The authors showed that the Amph I (1-278) fragment could cause both tau hyperphosphorylation and synaptic dysfunctions in Tau P301S mice. It is not clear in terms of the relationship between these two phenotypes – does one lead to the other or are both in parallel downstream of Amph I fragment? The authors should at least discuss different possibilities in the Discussion.

As suggested by the reviewer, we discussed this issue in the Discussion section (Page 17, Lines 366-373). Synaptic dysfunction and tau hyperphosphorylation are key pathological characteristics of AD. As discussed above, the Amphiphysin I (1-278) fragment may induce synaptic dysfunction and tau phosphorylation through independent pathways. It causes synaptic dysfunction by interfering with clathrin-mediated endocytosis and synaptic vesicle recycling, and enhances tau hyperphosphorylation through the p35/CDK5 pathway. However, it has been shown that tau hyperphosphorylation induces synaptic dysfunction. Thus, the hyperphosphorylated tau induced by Amphiphysin I (1-278) may also contribute to synaptic dysfunction.

10. Tau-P301S is associated with FTD, but not AD, although the Tau-P301S mouse model (PS19) is useful for tauopathy studies. The authors should discuss this clearly in the Discussion section.

We appreciate the reviewer’s comments. As suggested by the reviewer, we discussed this in the Discussion section (Page 17, Lines 360-365). Here we used the Tau P301S mice model to explore the role of Amphiphysin I fragments on tau phosphorylation and synaptic dysfunction. The Tau P301S mice overexpress a P301S mutant human tau that is indicated in frontotemporal dementia (FTD). Thus, the Tau P301S mice are associated with FTD. Since the mice develop tau pathology, synaptic dysfunction and behavioral impairments (Yoshiyama et al., 2007, Neuron 53(3): 337-351), it is widely used for the investigation of tauopathies. Thus, the results in the present study may reflect a common mechanism in tauopathies.

[Editors' note: further revisions were suggested prior to acceptance, as described below.]

Essential Revisions:1. In the Methods sections, a description of how the immunohistochemical images were quantified is still lacking. Please define "Optical density score", and indicate whether this refers to areas of tissue or within individual cells.

We have defined "Optical density score" in Methods sections. The images were analyzed by Image J, plus the IHC Profiler plugin. The program counts the pixels and evaluates the percentage contribution of high positive, positive, low positive, and negative. The "Optical density score" was calculated as (Percentage contribution of high positive*4+ Percentage contribution of positive*3+ Percentage contribution of low positive*2+ Percentage contribution of negative*1)/100.

2. In Figure 4D, AT8 and AT100 staining in the EGFP-Amph1 (279-695) group is markedly more intense than that of either EGFP-Vector or EGFP-Amph1 FL groups. However, the quantification graphs indicate that there is no difference between staining in these three groups. Why is this? Was the "Optical Density Score" calculated the same way as in Figure 2D and 2F?

The "Optical Density Score" calculated the same way as in Figure 2D and 2F. However, we calculated the optical density score of the entire image. The positive area of EGFP-Amph1 (279-695) group only accounted for a small percentage of the total area, which may explain why the difference between these groups were small.

3. In the legends for Figure 4 and 5, n's are listed as "n = 4" or n = 5". It seems that this should be corrected to "n = 4 mice per group" or "n = 5 mice per group", etc.

We have corrected these in legends.

4. The authors have added a western blot of ATP1A1 to illustrate the purity of their membrane preparations, in Figure 5. However, there is no description of this new data in the text or in the figure legend.

We have added the description in legend.

5. The authors have toned down a number of their statements to accurately reflect the data. However, the title for the legend for Figure 7 still needs to be revised. It reads "Amph1 N278A mutation ameliorates synaptic dysfunction and cognitive deficits in Tau P301S mice. However, since no data from wildtype mice is included in this figure, it is not clear that the P301S mice exhibited synaptic dysfunction and cognitive deficits. It would be more accurate to state something like "Expression of Amph1 N278A enhances synaptic and cognitive function in Tau P301S mice."

We appreciate the reviewer’s comments. We have revised the title for the legend for Figure 7.

6. A few typos still remain: Figure 6F X-axis label should read "Trials", not "Trails"; in Line 167, "loss" should be "lose". There are a number of minor grammatical errors throughout the text that could be easily corrected by a proof-reader.

We appreciate the reviewer’s comments. We have revised the errors throughout the text.

[Editors' note: further revisions were suggested prior to acceptance, as described below.]

The manuscript has been improved but there is one remaining issue that needs to be addressed, as described below (original comment and your response copied here for clarity):Comment 2. In Figure 4D, AT8 and AT100 staining in the EGFP-Amph1 (279-695) group is markedly more intense than that of either EGFP-Vector or EGFP-Amph1 FL groups. However, the quantification graphs indicate that there is no difference between staining in these three groups. Why is this? Was the "Optical Density Score" calculated the same way as in Figure 2D and 2F?"Your response: "The "Optical Density Score" calculated the same way as in Figure 2D and 2F. However, we calculated the optical density score of the entire image. The positive area of EGFP-Amph1 (279-695) group only accounted for a small percentage of the total area, which may explain why the difference between these groups were small."The images shown in Figure 4D show alterations in expression that are just as striking, or even more so, than those shown in Figure 2D and 2F. Yet, the accompanying graphs indicate that the expression levels between the groups in question in 4D are not different, whereas those in 2D and 2F are markedly and significantly different. There is an incongruency that needs to be resolved. This brings up important questions of how representative the images are, whether similar sized and similarly magnified images were quantified, and how robust your system of image selection and quantification is. Please either perform the quantification at the same magnifications, or provide images in Figure 4D that are more representative of the quantification results.

We are grateful to the constructive comment from the reviewer. We have provided images in Figure 4D that are more representative of the quantification results.